# `PersonalizedRouter`: Personalized LLM Routing via Graph-based User Preference Modeling

**Zhongjie Dai** [1,2]                                               *zhongjie@illinois.edu*
*University of Illinois at Urbana-Champaign*

**Tao Feng** [1]                                                     *taofeng2@illinois.edu*
*University of Illinois at Urbana-Champaign*

**Jiaxuan You**                                                      *jiaxuan@illinois.edu*
*University of Illinois at Urbana-Champaign*

**Reviewed on OpenReview:** *https://openreview.net/forum?id=W80eE3ArAl*

## Abstract

The growing number of Large Language Models (LLMs) with diverse capabilities and response styles provides users with a wider range of choices, which presents challenges in selecting appropriate LLMs, as user preferences vary in terms of performance, cost, and response style. Current LLM selection methods typically optimize for a single fixed objective, such as performance, cost, or a trade-off between them, and fail to learn individual user preferences from interaction data. To address these limitations, we propose `PersonalizedRouter`, a graph-based framework that models diverse user profiles and performs personalized LLM selection by leveraging interaction data that includes task context, queries, candidate LLMs, and user decisions. To capture contextual information between user queries and optimal LLMs, `PersonalizedRouter` converts the interaction data into a heterogeneous graph, where the relationships between different types of nodes are represented by edges. To evaluate adaptability across users, we design two strategies: the multi-cost-efficiency simulation strategy and the LLM-as-a-Judge strategy. In addition, we construct `PersonaRoute-Bench`, a large-scale benchmark with 1,000 simulated users and 10 LLMs. Experimental results show that `PersonalizedRouter` significantly outperforms existing LLM selection methods and surpasses the strongest methods by a large margin of 15.38% and 9.83% under two simulation strategies. On the `PersonaRoute-Bench` with 1,000 users, it further surpasses the best methods by 16.19% and 59.69% while maintaining higher efficiency. Moreover, `PersonalizedRouter` demonstrates strong few-shot generalization, achieving 64.81% and 85.80% of the fully trained model's performance when adapting to new users and new LLMs.

🐙 ulab-uiuc/PersonalizedRouter      🤗 Hugging Face Collection

## 1 Introduction

In recent years, the rapid growth of model scale and advances in training techniques have fueled the explosive emergence of LLMs (Feng et al., 2025c; 2024a; 2025a; Chen et al., 2025), offering users a diverse range of choices. Although large-scale language models have shown remarkable performance on many tasks, they tend to be inefficient when dealing with simple problems. In some scenarios, small-scale language models can achieve comparable performance while requiring fewer resources. Moreover, different LLMs excel at

---

[1]Equal contribution.
[2]Work was done while the author was an intern at the University of Illinois at Urbana Champaign. An additional contact email is dai020039@gmail.com

Table 1: **Comparison of `PersonalizedRouter` with current LLM selection methods.** Unlike previous approaches, `PersonalizedRouter` introduces an inductive graph framework for multi-user scenarios that leverages user interaction information, enabling it to capture latent user preferences to generalize to new users.

| Method | Multi-task Support | Generalization to New LLMs and Users | User preference modeling |
|---|---|---|---|
| Hybrid LLM (Ding et al., 2024) | ✗ | ✗ | ✗ |
| FrugalGPT (Chen et al., 2023) | ✗ | ✗ | ✗ |
| C2MAB-V (Dai et al., 2024) | ✗ | ✗ | ✗ |
| RouterDC (Shuhao et al., 2024) | ✗ | ✗ | ✗ |
| GraphRouter (Feng et al., 2024b) | ✓ | ✗ | ✗ |
| `PersonalizedRouter` | ✓ | ✓ | ✓ |

different tasks, exhibiting varying performance and cost efficiency on the specific application, while domain-specific expert models often achieve superior results on specialized tasks. In addition to differences in response quality and cost, LLMs also exhibit diverse response styles, which influence users' understanding of the query (Zhang et al., 2025b; Feng et al., 2024b). In multi-user scenarios, users often have distinct preferences that are difficult to model directly (Feng et al., 2025c; Zhang et al., 2025b; Sun et al., 2025), making it challenging for a single LLM to serve all users consistently. Therefore, this paper aims to draw attention to this pressing research question: *Given multiple user preferences, how can we design an LLM router that is personalized for each individual user?*

Current LLM selection methods aim to predict the most suitable LLM for a given user query using various strategies. HybridLLM (Ding et al., 2024) utilizes a pre-trained language model to make binary decisions between a large and a small LLM. FrugalGPT (Chen et al., 2023) uses a pre-trained model to score the responses to select the best LLM under a given cost constraint. RouterDC (Shuhao et al., 2024) encodes the input query using a pre-trained encoder and selects the LLM whose embedding is most similar to the query embedding. GraphRouter (Feng et al., 2024b) constructs a heterogeneous graph based on user interaction data and uses a Graph Neural Network (GNN) to predict the most suitable LLM for a given query. However, a fundamental limitation in current approaches is that they fail to adequately account for user preferences (Table 1). Many existing methods depend on simplistic representations such as BERT-based embeddings to distinguish between queries and optimize for a single fixed objective (e.g., performance only). This narrow focus restricts their ability to generalize to multi-user scenarios.

Existing methods struggle to adapt to multiple users because they fail to capture user preferences from historical interaction data and typically rely on fixed optimization objectives. To address these challenges, we propose `PersonalizedRouter`, a graph-based framework that leverages user interaction data to provide personalized LLM selection for different users. Specifically, `PersonalizedRouter` constructs a heterogeneous graph consisting of query nodes, task nodes, LLM nodes, and user nodes, and these nodes are connected through edges that represent different user preferences, such as performance-first or style-oriented choices. By aggregating information across different types of nodes, the GNN captures diverse latent user preferences through embeddings, enabling more effective LLM selection for new queries.

To comprehensively evaluate the adaptability of LLM selection methods in multi-user scenarios, we design two simulation strategies that model diverse user behaviors: multi-cost-efficiency, which calculates a reward score balancing performance and inference cost, and LLM-as-a-Judge, which leverages user profiles to instruct LLMs to simulate different preference groups. Based on these strategies, we construct `PersonaRoute-Bench`, a large-scale benchmark with over 1,000 simulated users and 10 LLMs, providing a realistic setting for evaluating scalability and user diversity. Experimental results show that `PersonalizedRouter` outperforms existing LLM selection methods by 9.83% and 15.38% under the two simulation settings. On `PersonaRoute-Bench`, it further surpasses the strongest baseline by 16.19% while maintaining high efficiency, with only a 5% drop in performance as the scale increases. To simulate dynamic real-world settings, we introduce new user and new LLM scenarios where interaction data from new users or new LLMs are excluded during training. `PersonalizedRouter` achieves 64.81% and 85.90% of the fully trained model's performance in these settings, respectively. To mitigate potential biases introduced by simulated user preferences, we additionally collect

a small-scale human interaction dataset. Our model outperforms the best baseline by 6.05%, demonstrating strong generalization to real-user scenarios.

To summarize, our main contributions are as follows.

- We propose `PersonalizedRouter`, a graph-based personalized routing framework for multi-user LLM selection problems that models diverse user preferences and generalizes effectively to unseen users and new LLMs.

- We propose two simulation strategies to evaluate the adaptability of methods, considering response quality, inference cost, and response style. Our model consistently outperforms baseline models by at least 9.83%. Furthermore, it surpasses all baselines not only in the large-scale setting with 1,000 simulated users, but also in the small-scale human interaction setting.

- We construct `PersonaRoute-Bench`, a large-scale benchmark with over 1,000 simulated users and 10 LLMs, designed to emulate real-world personalized routing scenarios and evaluate scalability, generalization, and robustness of multi-user LLM selection methods.

## 2 Related Works

### 2.1 LLM Selection

With the emergence of LLMs with diverse model scales, users now have the option to choose not only high-performance but high-cost models, but also smaller LLMs that offer competitive performance. This scenario motivates various LLM selection strategies. From the cost-efficiency perspective, Zhu et al. (2023) fine-tunes a pre-trained language model to predict the appropriate LLM to achieve lower overall computational cost. Ding et al. (2024) considers not only cost but also response quality, aiming to select small-scale LLM whenever the quality difference is within an acceptable range. RouterLLM (Ong et al., 2024) leverages real user preference data from the Chatbot Arena and applies matrix factorization for query assignment. Beyond binary LLM selection, other approaches extend the setting to multiple candidate LLMs. Chen et al. (2023) scores all responses and selects the one with the highest score under the given cost budget. PolyRouter (Stripelis et al., 2024) explores routing strategies built upon k-nearest neighbors and a multilayer perceptron. Feng et al. (2024b) formulates the LLM selection task as a link prediction problem, predicting the link between a query and the best LLM. C2MAB-V (Dai et al., 2024) employs a bandit-based routing model with an exploration mechanism to balance exploration and exploitation when selecting LLMs. RouterDC (Shuhao et al., 2024) selects the most suitable LLM based on the cosine similarity between the query and each LLM's embedding. Building upon existing LLM selection methods, we introduce a multi-user scenario and design two distinct simulation strategies to model user preference. By leveraging the user interaction data through GNN, our framework can effectively generalize to new users without retraining.

### 2.2 GNN for Link Prediction

GNNs are a class of neural networks designed to learn node embeddings by aggregating information from neighboring nodes, including GCN (Kipf & Welling, 2017), GraphSAGE (Hamilton et al., 2017), and GAT (Veličković et al., 2017). Based on these foundations, heterogeneous GNNs, including HeterGNNs (Hu et al., 2020; Peng et al., 2019; Schlichtkrull et al., 2017) and HGATs (Wang et al., 2019) have been developed to handle graphs containing multiple types of nodes and edges. Furthermore, GNNs have demonstrated strong zero-shot and few-shot generalization capabilities, making them well-suited for tasks with limited supervision (Gao & Xu, 2020; Fey et al., 2023; Cao et al., 2023; Chen et al., 2022). The strong embedding capabilities of GNNs have led to significant advancements in applications, including recommender systems (Min et al., 2022) and social network analysis (Wu et al., 2020). Link prediction is an important GNN application, aiming to infer potential connections between nodes from the existing graph structure. It also plays an important role in areas such as bioinformatics (Zitnik et al., 2018; Long et al., 2022) and recommender systems (Wu et al., 2021; He et al., 2020b). In the static graph setting, node representation-based GNN approaches (Huang et al., 2023; Wang et al., 2022; Feng et al., 2025b; Yu et al., 2024; Zhang et al., 2025a;

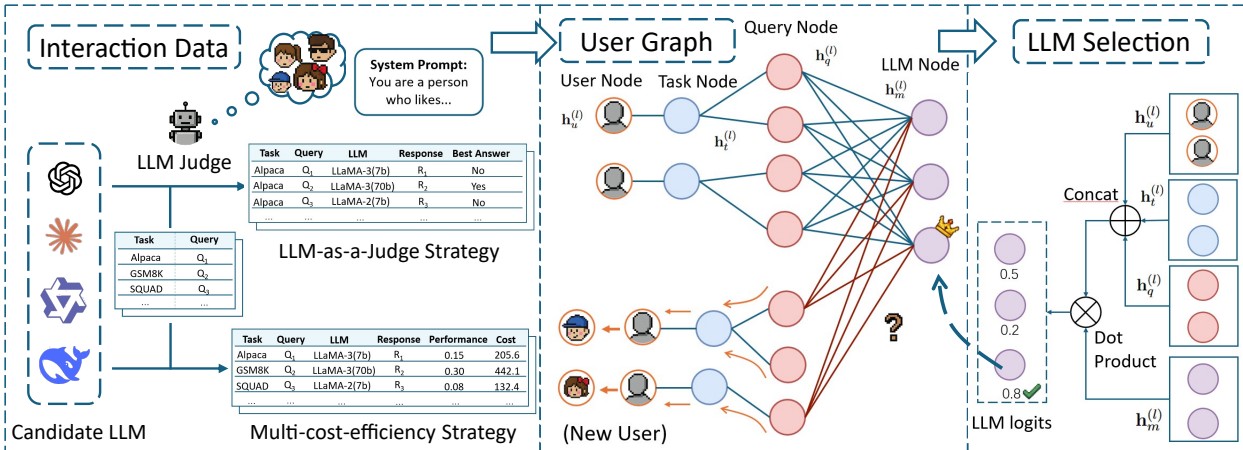

Figure 1: **Overview of `PersonalizedRouter` methodology**. As shown in the left part, we first utilize the candidate LLMs to generate responses based on the multi-task dataset. Next, under two simulation strategies, we obtain the corresponding interaction data. As illustrated in the middle part, `PersonalizedRouter` transforms the user interaction data into a graph, where nodes represent the user, task, query, and LLM, and the edges capture the relationships between different node types. In the right part, we leverage a GNN to embed both node and edge features, updating and capturing the user's hidden features. Ultimately, we select the optimal LLM from the predicted probability distribution.

Feng et al., 2023) and local subgraph-based methods (Chamberlain et al., 2023; Yun et al., 2022) are capable of handling more complex scenarios. Inspired by these works, we apply GNNs to the LLM selection problem, aiming to capture users' latent preferences and make more effective LLM predictions.

## 3 Methods

In this section, we present `PersonalizedRouter`, a graph-based framework for LLM selection based on user interaction data. An overview is illustrated in Figure 1. We introduced the problem of user-level LLM selection (Section 3.1). Next, we present the `PersonalizedRouter` framework, which is capable of generalizing for diverse users (Section 3.2).

### 3.1 Problem Formulation

For the LLM selection problem, the router serves as a critical component that is responsible for understanding user requirements and preferences to make optimal selections. Given a user query $q_u$ that contains contextual information, the router selects the most suitable LLM $M_i$ from the given candidate pool $\{M_1, M_2, \ldots, M_n\}$, aiming to optimize multiple factors, including model performance, computational cost, and answer styles. Since the user profile is not directly accessible, the router is trained on user interaction data containing query content, response quality, and answer feedback, which is expected to allow the router to capture hidden user preferences and generalize effectively on diverse users.

### 3.2 `PersonalizedRouter` Framework

**Users Graph Construction.** We leverage user interaction data to construct a user graph, where the GNN learns to represent hidden user preferences through message passing. To construct the graph, the framework extracts users, tasks, queries, and LLMs from the interaction data and represents them as different types of nodes in a heterogeneous graph. Specifically, queries represent the users' contextual information, while tasks correspond to the task type of each query. Edges are constructed based on the relationships among these entities, including user–task, task–query, and query–LLM. With the constructed graph, we apply a GNN to

embed node and edge features, enabling message passing across the graph and model user preferences, and constructing user profiles for LLM selection.

**Initialization of Node Features.** In the graph constructed by `PersonalizedRouter`, we define four types of nodes: user nodes $h_u^{(l)}$, task nodes $h_t^{(l)}$, query nodes $h_q^{(l)}$, and LLM nodes $h_m^{(l)}$. Correspondingly, we construct three types of edges: user–task edge, task–query edge, and query–LLM edge. For node initialization, the textual descriptions of each entity are encoded using a shared pre-trained language model (e.g., BERT (Devlin, 2018)), and the resulting embeddings are used as the initial node representations. The detailed descriptions of tasks and LLMs are provided in Appendix A.5. For the user nodes, we initialize their embeddings using one-hot encodings based on user interaction. For edge initialization, user–task and task–query edges are assigned an initial weight of 1. In contrast, query–LLM edges are initialized differently depending on the simulation setting. For the multi-cost-efficiency simulation strategy setting, the edge features are derived from interaction data, incorporating a combination of performance and cost. Under the LLM-as-a-Judge strategy setting, the edge embedding is initialized as a binary indicator, reflecting whether the LLM produced the best response according to the LLM judge.

**Heterogeneous GNN.** Based on the constructed user graph, we utilize a heterogeneous GNN as the prediction model $f$, which aggregates information from four types of nodes (user, task, query, and LLM nodes) and three types of edges (LLM–user-task edges, task–query edges, and LLM–query edges). The heterogeneous GNN performs multiple rounds of message passing and weighted aggregation to extract information from the local neighborhoods to capture meaningful node embeddings. Therefore, even in the absence of explicit user profiles, the user embeddings can be inferred through GNN iterations, which enables more effective predictions. In our framework, for an l-layer GNN, the node embedding at the l-th layer is updated as:

$$\mathbf{h}_u^{(l)} = \mathbf{H}_u^{(l)}\Big(\text{MEAN}\Big(\sigma(\mathbf{w}_{ut}^T\mathbf{W}_u^{(l)}\mathbf{h}_n^{(l-1)})\Big) \oplus \mathbf{h}_u^{(l-1)}\Big), \tag{1}$$

$$\mathbf{h}_t^{(l)} = \mathbf{H}_t^{(l)}\Big(\text{MEAN}\Big(\sigma(\mathbf{w}_1\mathbf{W}_1^{(l)}\mathbf{h}_n^{(l-1)})\Big) \oplus \mathbf{h}_t^{(l-1)}\Big), \tag{2}$$

$$\mathbf{h}_q^{(l)} = \mathbf{H}_q^{(l)}\Big(\text{MEAN}\Big(\sigma(\mathbf{w}_1\mathbf{W}_1^{(l)}\mathbf{h}_n^{(l-1)})\Big) \oplus \mathbf{h}_q^{(l-1)}\Big), \tag{3}$$

$$\mathbf{h}_m^{(l)} = \mathbf{H}_m^{(l)}\Big(\text{MEAN}\Big(\sigma(\mathbf{w}_{mq}^T\mathbf{W}_m^{(l)}\mathbf{h}_n^{(l-1)})\Big) \oplus \mathbf{h}_m^{(l-1)}\Big). \tag{4}$$

where $\mathbf{h}_u^{(l)}, \mathbf{h}_t^{(l)}, \mathbf{h}_q^{(l)},$ and $\mathbf{h}_m^{(l)}$ represent the node embedding after $l$ layers of message passing of the user, task, query, and LLM nodes, respectively. The four node embeddings are initialized as $\mathbf{h}_u^{(0)}, \mathbf{h}_t^{(0)}, \mathbf{h}_q^{(0)}, \mathbf{h}_m^{(0)} = e_u, e_t, e_q, e_m$, respectively. $n \in \mathcal{N}(v)$ denotes the neighboring nodes of $v$, and $v$ can be a task, query, user, or LLM node. $\sigma(\cdot)$ is the activation function, such as ReLU or softmax, and we utilize ReLU. $\oplus$ denotes vector concatenation. $\mathbf{1}[v \in V_d, u \in V_t]$ represents the distinct message type. For task nodes, the message is from user to task or from query to task, and for query nodes, the message is from task to query or from LLM to query. In addition, $\mathbf{w}_{\mathbf{1}[t \in V_t, m \in V_t]}$ indicates that different edge types correspond to different edge weights; specifically, for query nodes, if it is from task to query, it is represented as $w_{tq}$, and from LLM to query, it is represented as $w_{mq}$, and for task nodes, if it is from user to task, it is represented as $w_{ut}$, and from query to task, it is represented as $w_{qt}$. $\mathbf{H}^{(l)}, \mathbf{W}^{(l)}$ are learnable parameters.

We formulate the LLM selection problem as the link prediction problem in a constructed user graph, where the model is trained to predict the probability of the edge between the given query and each candidate LLM and set edge labels for training based on the different simulation strategies. Under the multi-cost-efficiency simulation strategy, the best LLM is identified with the highest reward score associated with the trade-off between accuracy and inference cost. Under the LLM-as-a-Judge strategy, an LLM acting as a judge simulates user preferences and determines the optimal LLM based on the content of the responses. Consequently, we assign an edge label of 1 to the best LLM and 0 to all remaining query–LLM edges. After multiple GNN iterations, we obtain final representations of four types of nodes. We first combine the embeddings of the user, task, and query nodes to generate a unified embedding that jointly captures information from all three aspects $\mathbf{h}_{uqt}^{(l)} = \text{MLP}(\text{CONCAT}(\mathbf{h}_u^{(l)}, \mathbf{h}_t^{(l)}, \mathbf{h}_q^{(l)}))$. Next, we predict the probability for each edge using EdgePred$(\cdot)$, which is formulated as $\hat{y}_{logits} = \text{MEAN}\Big(\text{DOT}(\mathbf{h}_{uqt}^{(l)}, \mathbf{h}_m^{(l)})\Big)$. Ultimately, we identify the most suitable one by selecting the edge with the highest scores: $\hat{y} = \arg\max_m \Big(\text{EdgePred}(h_{uqt}, h_m)\Big)$.

Table 2: **Overview of the task datasets.** We sampled 600 cases from each of four distinct types of task datasets. Each task dataset is associated with a different evaluation metric.

| Dataset | Task Type | Metric | Cases |
|---------|-----------|--------|-------|
| Alpaca | Hybrid QA | F1 | 600 |
| GSM8K | Multi-step Reasoning | Accuracy | 600 |
| SQuAD | Reading Comprehension | F1 | 600 |
| Multi-News | Article Summary | F1 | 600 |

**Generalization to New Users and New LLMs.** Existing LLM selection methods often struggle to represent diverse user scenarios, as they are typically built upon simplistic, fixed objectives that constrain either performance or cost. Such rigid assumptions limit their effectiveness in real-world applications, where systems must adapt to varying user needs. Furthermore, as companies rapidly iterate on their LLMs, it becomes increasingly important for the router to remain both effective and robust when dealing with new models. To evaluate the real-world potential of our model and baselines, we construct an auxiliary dataset under both simulation strategies following (Cao et al., 2023; Fey et al., 2023), which includes query-level interaction records from new users or new LLMs, sampled from the same distribution as the training data. We train all models using interaction data from known users or known LLMs only, while the auxiliary dataset is excluded from the training phase and is instead used during testing to initialize the GNN in a few-shot setting, allowing us to assess the model's ability to generalize to unseen users or LLMs.

## 4 Experimental Setup

### 4.1 Candidate LLMs and Task Datasets

We selected a set of candidate LLMs for our experiments, using the Together AI, OpenRouter AI, and NVIDIA Build. Details of these models are provided in Appendix A.6, including their sizes and cost per million tokens.

In this paper, we focus on modeling the relationship between user queries and LLM answers. To simulate typical user input scenarios in real-world settings, we select four representative datasets. For user daily chat behavior, we use **Alpaca** (Taori et al., 2023), a hybrid question-answering (QA) dataset containing 52K samples, which covers a wide range of tasks such as casual conversation and instruction following, effectively simulating user queries in daily scenarios. For multi-step mathematical reasoning, we adopt **GSM8K** (Cobbe et al., 2021), which includes 8.5K school math word problems that require multi-hop reasoning. For contextual understanding, we use **SQuAD** (Rajpurkar, 2016), a widely-used QA dataset consisting of over 100K question-answer pairs linked to more than 500 Wikipedia articles. For information details, we choose **Multi-News** (Fabbri et al., 2019), a multi-document summarization dataset consisting of 56K news–summary pairs written by professional editors, reflecting the capability for capturing key information. The task types, evaluation metrics, and the number of selected tasks from four task datasets are detailed in Table 2.

### 4.2 Construction of Interaction Datasets

**Two Simulation Strategies.** Current LLM selection methods typically rely on fixed cost-efficiency constraints, e.g., cost-first or performance-first preferences, which limit the adaptability to diverse user scenarios. Therefore, we introduce the first simulation strategy: multi-cost-efficiency simulation strategy, which considers a set of cost-efficiency constraints at the same time to measure how effectively the method adapts to various user preferences. With the emergence of models that emphasize emotional expression, users are increasingly valuing not only the performance of LLMs but also the response styles. Recent studies have shown that the system prompt can influence the persona exhibited by an LLM (Zhong et al., 2024; Kong et al., 2024). Therefore, we propose the second simulation strategy, LLM-as-a-Judge strategy, in which we employ

additional LLMs as judges, using system prompts constructed from user profiles to simulate different users. This approach evaluates the router's ability to generalize across users with diverse contextual expectations.

Under the two simulation strategies, using the four task datasets introduced in Section 4.1 and a pool of candidate LLMs, we construct two interaction datasets. First, we uniformly sample queries from four task datasets and merge them into a query set. Then, for each query, we collect responses from all candidate LLMs based on the two simulation strategies with different metrics to build interaction datasets. The structure of the two interaction datasets is shown on the left side of Figure 1.

For the **multi-cost-efficiency simulation strategy**, we collect responses for each query from candidate LLMs, which contain performance, cost, and reward value. Methods are required for optimal routing decisions that maximize the reward score.

- **Performance** value is to evaluate the quality of the LLM's response using different task metrics mentioned in Section 4.1

- **Cost** value is measured with the total number of tokens calculated by GPT-2 and the corresponding token cost for each LLM (Appendix A.6).

- **Reward** value reflects the trade-off between performance and cost. To ensure comparability, we first normalize both performance and cost. Next, we define $Reward = \alpha \cdot Performance - \beta \cdot Cost$. To simulate diverse user preferences, we introduce nine $\alpha$ and $\beta$ weight pairs (Appendix A.1), representing different user types ranging from those who prioritize high performance to those who prefer cost-efficient performance.

For the **LLM-as-a-Judge strategy**, we first collect responses for each query from candidate LLMs and then utilize an additional LLM as a judge to generate a binary label called best-answer based on distinct user profiles. The models used as judges are shown in Appendix A.2, the detailed instruction prompts can be found in AppendixA.3, and the user profiles are listed in Appendix A.4

- **Best Answer** is selected by the LLM judge, which represents the response that best aligns with the predefined user profiles. The label reflects users' preferences on diverse response styles in the real scenario.

**Datasets Splitting.**  After generating the corresponding interaction datasets under the two simulation strategies, each interaction dataset is applied to two experimental settings: a standard setting and a new user setting. For both settings, the dataset is divided into three parts, training, validation, and test sets with a ratio of $70\% : 10\% : 20\%$. In the standard setting, all user interaction data is accessible, allowing the model to learn user preferences from historical data and capture hidden user profiles more effectively. In the new user setting, we assume that the first three users are new users, while the remaining six users are visible, whose interaction data is available for model training. As mentioned in Section 3.2, we remove all interaction data of new users from the training and validation sets, while keeping the test set unchanged. Next, we construct an auxiliary dataset following (Cao et al., 2023; Fey et al., 2023), which consists of a uniformly sampled query subset of new users in the training set. This auxiliary dataset is used as a few-shot dataset only during the testing phase for the model to adapt to unseen users.

### 4.3  Baseline

In this paper, we introduce the following baselines to compare with `PersonalizedRouter`.

- **Hybrid LLM** (Ding et al., 2024) is designed for scenarios with only two LLMs. It trains a pre-trained language model to assign queries to either a small or a large LLM, aiming to balance various factors such as performance and cost. In the experiment, we replace DeBERTa (He et al., 2020a) with RoBERTa (Liu, 2019) as the pre-trained model for routing queries, which shows better performance.

- **FrugalGPT** (Chen et al., 2023)A pre-trained language model is used to generate scores for responses generated by candidate LLMs, providing an evaluation of response quality. Under a given total cost, the LLM with the highest score is selected as the final executor. In the experiment, we use RoBERTa (Liu, 2019) as the pre-trained model.

- **RouterDC** (Shuhao et al., 2024) uses a pre-trained model mDeBERTaV3-base (He et al., 2023) to encode the input query, and computes cosine similarity with each candidate LLM embedding. The LLM with the highest score is selected as the final prediction.

- **GraphRouter** (Feng et al., 2024b) models user interaction data as a heterogeneous graph, and uses a GNN to learn the relations between queries and LLMs, and selects the LLM based on the scores of query–LLM edges.

Furthermore, to provide a more comprehensive evaluation of `PersonalizedRouter`, we incorporate the optimal solution as a golden baseline.

- **Oracle** represents the upper bound achieved by the best selection, where each query is routed to the most suitable LLM.

### 4.4 Implementation Details

For router training, we use a two-layer graph attention network with a hidden dimension of 32. The model is trained with a batch size of 32 for up to 400 epochs. We use the Adam optimizer (Kingma & Ba, 2014) and apply a LambdaLR scheduler to gradually decay the learning rate from 1e-3 to 0 during training. Our method is implemented using PyTorch and PyG, and all experiments are conducted on an NVIDIA A6000 48GB Tensor Core GPU. In terms of LLMs, we use Together AI, OpenRouter AI, and NVIDIA Build for calling candidate LLMs and an LLM judge for the response.

## 5 Experimental Results

### 5.1 Comparison with Existing Baseline

We compare `PersonalizedRouter` with other representative baseline methods under two different simulation strategies. All models are trained under the general experimental setting, which involves 10 LLMs and 9 users, aiming to evaluate their ability to adapt to new queries from existing users. Detailed information about the LLMs and users is provided in Appendix A.5 and Appendix A.4, respectively.

**Multi-cost-efficiency Simulation Strategy.** Under the multi-cost-efficiency simulation strategy, the reward score represents the trade-off between performance and cost (Section 4.2). As shown in Table 3, `PersonalizedRouter` consistently outperforms all baseline methods and surpasses the strongest methods by a large margin of 15.38%. Furthermore, `PersonalizedRouter` achieves 83.88% of the oracle performance, demonstrating its strong adaptability and effective selection strategy.

**LLM-as-a-Judge Strategy.** Under the LLM-as-a-Judge strategy, the metric accuracy measures the prediction capability on new queries based on interaction data (Section 4.2). As shown in Table 3, `PersonalizedRouter` shows better performance than other methods in terms of accuracy. Compared to the best baseline, `PersonalizedRouter` achieves a 9.83% advantage, further demonstrating its effectiveness.

### 5.2 Comparison with Baselines at Larger Scale

To further assess the scalability of the router, we constructed `PersonaRoute-Bench`, a large-scale benchmark with over 1,000 simulated users and 10 LLMs for real-world routing evaluation. Under the multi-cost-efficiency simulation strategy, we evaluated 10 LLMs across 1,000 users. Under the LLM-as-a-Judge strategy, to further mitigate the bias of relying on a single LLM as a judge, we considered three different LLMs, and for each model, we applied two types of instruction prompts, resulting in six distinct judge configurations.

Table 3: **Comparison between different methods under two simulation strategies.** The Improvement is measured relative to the best baseline. The best result for each routing method is highlighted in **bold** and the second best result is highlighted with an underline.

| Scenario | Multi-cost-efficiency Simulation | | Scenario | LLM-as-a-Judge | |
|---|---|---|---|---|---|
| Method | Reward | Improvement (%) | Method | Accuracy | Improvement (%) |
| HybridLLM | 0.141 | -36.20 | HybridLLM | 0.347 | -14.74 |
| FrugalGPT | 0.218 | -1.36 | FrugalGPT | 0.354 | -13.22 |
| GraphRouter | 0.221 | 0.00 | GraphRouter | 0.364 | -10.57 |
| RouterDC | 0.208 | -5.88 | RouterDC | 0.407 | 0.00 |
| PersonalizedRouter | **0.255** | 15.38 | PersonalizedRouter | **0.447** | 9.83 |
| Oracle | 0.304 | 37.56 | Oracle | 1.000 | 145.70 |

Table 4: **Comparison between different methods under two simulation strategies in the large-scale experimental setting.** The Deltas represent the improvement of the Reward over the best baseline. The Reduction indicates the relative reduction in time cost compared to the baseline with the highest time cost. The best result for each routing method is highlighted in **bold** and the second best result is highlighted with an underline. The oracle represents the best selection without a defined time cost, which we indicate using slashes (/).

| Scenario | Multi-cost-efficiency Simulation | | | | LLM-as-a-Judge | | | |
|---|---|---|---|---|---|---|---|---|
| Method | Reward | Delta (%) | Time | Reduction (%) | Accuracy | Delta (%) | Time | Reduction (%) |
| HybridLLM | 0.119 | -43.33 | **4:32** | 98.50 | 0.196 | 0.00 | **4:56** | 98.40 |
| FrugalGPT | 0.141 | -32.86 | 47:44 | 84.29 | 0.108 | -44.90 | 49:31 | 83.92 |
| GraphRouter | 0.204 | -2.86 | 8:26 | 97.22 | 0.137 | -30.10 | 9:28 | 96.93 |
| RouterDC | 0.210 | 0.00 | 303:47 | 0.00 | 0.153 | -21.94 | 308:02 | 0.00 |
| PersonalizedRouter | **0.244** | 16.19 | 10:15 | 96.63 | **0.313** | 59.69 | 11:37 | 96.23 |
| Oracle | 0.310 | 47.62 | / | / | 1.000 | 410.20 | / | / |

For each configuration, 200 users were simulated, leading to a total of 1,200 simulated user preferences. In other words, each user profile is evaluated under six distinct judge configurations, effectively yielding six different user perspectives. Ultimately, the combination of 200 user profiles with six judge configurations results in a total of 1,200 simulated user samples. Details of the user profiles used for simulated user profiles are provided in Appendix A.4, and the experimental results are shown in the Table 4. Consistent with the small-scale results in Section 5.1, our model outperforms all baselines under both simulation strategies. Specifically, under the two simulation strategies, our model outperforms the strongest baseline by 16.19% and 59.69%, respectively. Moreover, the results indicate that our model achieves better performance while requiring less computation time.

To more intuitively demonstrate the scalability of `PersonalizedRouter`, we compare the results from two experiments conducted at different scales. The results provided in Table 5 indicate that `PersonalizedRouter` exhibits stable performance, with 5.2% drop relative to the Oracle.

## 5.3 Generalization to New Users

To evaluate the ability of different LLM selection methods to generalize to new users, we also train all models under the new user experimental setting (Section 3.2). Specifically, we treat the first three users as new users and the remaining six as known users for model training. Detailed user information is provided in Appendix A.4. To ensure consistent evaluation standards, we append the same auxiliary dataset to the test sets of all baseline methods. The final results are presented in Table 6.

Table 5: **Comparison between different methods under two simulation strategies on two experiment scales.** The small scale refers to the setting in the paper with 10 LLMs and 9 users, while the large scale refers to the supplementary setting with 10 LLMs and 1000 users. The Ratio is relative to the Oracle. The Δ represents the difference between the small-scale and large-scale ratios. The best result for each routing method is highlighted in **bold** and the second best result is highlighted with an underline.

| Scenario | Multi-cost-efficiency Simulation | | | | | LLM-as-a-Judge | | | | |
|---|---|---|---|---|---|---|---|---|---|---|
| Scale | Small Scale | | Large Scale | | | Small Scale | | Large Scale | | |
| Method | Reward | Ratio | Reward | Ratio | Δ | Accuracy | Ratio | Accuracy | Ratio | Δ |
| HybridLLM | 0.141 | 0.464 | 0.119 | 0.384 | 0.080 | 0.347 | 0.347 | 0.196 | 0.196 | 0.151 |
| FrugalGPT | 0.218 | 0.717 | 0.141 | 0.455 | 0.262 | 0.354 | 0.354 | 0.108 | 0.108 | 0.246 |
| GraphRouter | 0.221 | 0.726 | 0.204 | 0.658 | 0.068 | 0.364 | 0.364 | 0.137 | 0.137 | 0.227 |
| RouterDC | 0.208 | 0.684 | 0.210 | 0.677 | **0.007** | 0.407 | 0.407 | 0.153 | 0.153 | 0.254 |
| PersonalizedRouter | **0.255** | **0.839** | **0.244** | **0.787** | 0.052 | **0.447** | **0.447** | **0.313** | **0.313** | **0.134** |
| Oracle | 0.304 | 1.000 | 0.310 | 1.000 | 0.000 | 1.000 | 1.000 | 1.000 | 1.000 | 0.000 |

Table 6: **Comparison between different methods under simulation strategies in the new user experimental setting.** Improvement is measured relative to the best baseline. The best result for each routing method is highlighted in **bold** and the second best result is highlighted with an underline. Few-shots denotes PersonalizedRouter (few-shots), and Trained denotes PersonalizedRouter (trained).

| Scenario | Multi-cost-efficiency Simulation | | Scenario | LLM-as-a-Judge | |
|---|---|---|---|---|---|
| Method | Reward | Improvement (%) | Method | Accuracy | Improvement (%) |
| HybridLLM | -0.142 | -240.59 | HybridLLM | 0.294 | 0.00 |
| FrugalGPT | 0.044 | -56.44 | FrugalGPT | 0.192 | -34.69 |
| GraphRouter | 0.083 | -17.82 | GraphRouter | 0.256 | -12.93 |
| RouterDC | 0.101 | 0.00 | RouterDC | 0.208 | -29.25 |
| **Few-shots** | 0.07 | -30.69 | **Few-shots** | 0.313 | 6.46 |
| **Trained** | **0.108** | 6.93 | **Trained** | **0.326** | 10.88 |
| Oracle | 0.116 | 14.85 | Oracle | 1.000 | 240.14 |

**Multi-cost-efficiency Simulation Strategy.** Under the multi-cost-efficiency simulation strategy, the router makes trade-offs between performance and cost to achieve the best reward score without access to explicit user preferences. As shown in Table 6, PersonalizedRouter (few-shots) achieves 69.30% of the performance of the best baseline. Moreover, despite a limited few-shot interaction data from new users, PersonalizedRouter (few-shots) achieves 71.30% of PersonalizedRouter (trained), which demonstrates the strong generalization ability of our framework to new users.

**LLM-as-a-Judge Strategy.** Under the LLM-as-a-Judge strategy with an auxiliary dataset, the method demonstrates strong performance on new users. PersonalizedRouter (few-shots) improves 6.46% performance over the best-performing baseline and achieves 96.01% of its trained model, which demonstrates the scalability and efficiency of our framework in handling new users with few-shot supervision.

### 5.4 Generalization to New LLMs

To evaluate the generalization capability of PersonalizedRouter to new LLMs, we conduct experiments under the new LLM experimental setting (Section 3.2). Similar to the generalization to new users' settings, the model is trained on data from the first 10 LLMs, while the remaining 5 LLMs are treated as an auxiliary dataset for evaluation. Detailed information about the LLMs is provided in Appendix A.5. The final results are presented in Table 7.

Table 7: **Comparison between different methods under simulation strategies in the new LLM experimental setting.** The Improvement is measured relative to the best baseline. The best result for each routing method is highlighted in **bold** and the second best result is highlighted with an underline. Few-shots denotes `PersonalizedRouter` (few-shots), and Trained denotes `PersonalizedRouter` (trained).

| Scenario | Multi-cost-efficiency Simulation | | Scenario | LLM-as-a-Judge | |
|----------|----------|----------|----------|----------|----------|
| Method | Reward | Improvement (%) | Method | Accuracy | Improvement (%) |
| HybridLLM | 0.137 | -37.16 | HybridLLM | 0.273 | 0.00 |
| FrugalGPT | 0.182 | -16.51 | FrugalGPT | 0.038 | -86.08 |
| GraphRouter | 0.203 | -6.88 | GraphRouter | 0.247 | -9.52 |
| RouterDC | 0.218 | 0.00 | RouterDC | 0.199 | -27.10 |
| **Few-shots** | 0.201 | -7.80 | **Few-shots** | 0.329 | 20.51 |
| **Trained** | **0.234** | 7.34 | **Trained** | **0.383** | 40.29 |
| Oracle | 0.371 | 70.18 | Oracle | 1.000 | 266.30 |

Table 8: **Comparison between different methods under small-scale human interaction dataset.** The Improvement is measured relative to the best baseline. The best result for each routing method is highlighted in **bold** and the second best result is highlighted with an underline.

| Scenario | Human-as-a-Judge | |
|----------|----------|----------|
| Method | Accuracy | Improvement (%) |
| HybridLLM | 0.315 | -23.73 |
| FrugalGPT | 0.350 | -15.25 |
| GraphRouter | 0.357 | -13.56 |
| RouterDC | 0.413 | 0.00 |
| `PersonalizedRouter` | **0.438** | 6.05 |
| Oracle | 1.000 | 142.13 |

**Multi-cost-efficiency Simulation Strategy.** Under the multi-cost-efficiency simulation strategy, `PersonalizedRouter` (few-shots) performs closely to the best baseline with a 7.80% gap. It also achieves 85.90% of the performance of `PersonalizedRouter` (trained), demonstrating strong generalization capabilities.

**LLM-as-a-Judge Strategy.** Under the LLM-as-a-Judge strategy, `PersonalizedRouter` (few-shots) outperforms all baselines and achieves 85.90% of the performance of `PersonalizedRouter` (trained), demonstrating that our model remains robust and effective when faced with new LLMs.

## 5.5 Comparison with Baselines on Real-User Dataset

Although simulated users can partially validate whether the router is capable of modeling latent user preferences, they still differ from real users with diverse preferences. To further verify whether the router is effective in multiple real-user settings, we design a Human-as-a-Judge scenario, in which we collect a small-scale human interaction dataset for experimentation. Similar to the LLM-as-a-Judge scenario, in the Human-as-a-Judge setting, we recruited 40 users and provided them with 80 queries (20 queries selected from each of the four datasets in Section 4.1), each answered by 10 LLMs. Users were asked to select the single response that best matched their personal preference, and their selections were used to generate a binary label called best-answer. The questionnaire template is provided in Appendix A.9. Accordingly, the router must predict the response that aligns with user preferences on this real-user dataset. The experimental results are reported in Table 8. The results show that `PersonalizedRouter` achieves strong accuracy on the small-scale real-user dataset, demonstrating good generalization beyond simulated data to real-user settings.

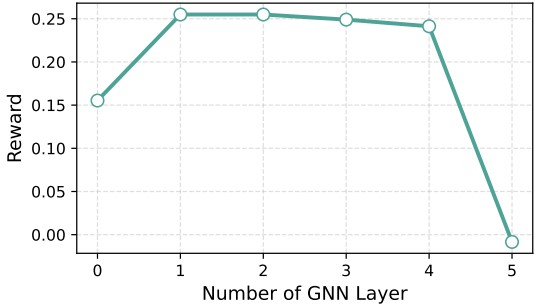

(a) Reward of different numbers of GNN layers under the multi-cost-efficiency simulation strategy.

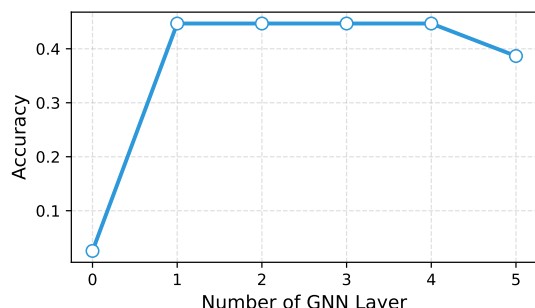

(b) Accuracy of different numbers of GNN layers under the LLM-as-a-Judge strategy.

Figure 2: **Comparison of reward and accuracy under different GNN layer counts using two simulation strategies.**

### 5.6 Ablation Studies

**The impact of GNN depth on `PersonalizedRouter`'s prediction performance.** Under two simulation strategies, we further explore the impact of the GNN layer on prediction performance. As shown in Figure 2a and 2b, we evaluate models with GNN layers ranging from 0 to 5. The results show that prediction performance improves with increasing GNN layers, peaking at 2 or 3 layers, but begins to decline as the GNN becomes deeper. We believe that GNN helps aggregate information across different types of data, but excessive depth leads to over-smoothing, where node representations become increasingly similar, ultimately degrading the model's predictive performance.

## 6 Conclusion

In this paper, we propose `PersonalizedRouter`, a graph-based inductive framework for multi-user LLM selection. Unlike existing methods that ignore user diversity and lack access to explicit user profiles, our approach models user preferences as latent representations learned from interaction data. We formulate the selection process as a link prediction task, in which GNN predicts the probability scores of edges between queries and each candidate LLM. To evaluate adaptability to diverse users, we introduce two simulation strategies: multi-cost-efficiency, which aims to evaluate the trade-off between performance and cost, and LLM-as-a-Judge, which utilizes an additional LLM guided by diverse user profiles to select the best answer.Furthermore, we construct `PersonaRoute-Bench`, a large-scale benchmark comprising over 1,000 simulated users and 10 candidate LLM. Experimental results show that `PersonalizedRouter` consistently outperforms competitive baselines across all settings, including new-user and new-LLM scenarios, demonstrating strong generalization ability. With the explosive emergence of various LLMs, performance is no longer the only factor to consider. Therefore, personalized LLM selection based on individual preferences has become an important and practical research topic. We hope this paper will facilitate more user-centric LLM routing research.

## 7 Limitations

This paper proposes two simulation strategies to evaluate whether LLM selection methods can be generalized to new users. However, in real-world applications, user behavior is often more complex, potentially involving a mixture of preferences and evolving over time. Future work will explore more sophisticated ways to represent and learn dynamic and diverse user preferences. For stronger theoretical grounding, future research may explore augmenting the current model with an explicit utility function, thereby better achieving the optimization objective.

## Acknowledgments

We sincerely appreciate the research gift from Lenovo that made this project possible.

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

# A Appendix

## A.1 Weight pairs of calculating reward metric

Under the multi-cost-efficiency simulation strategy, we define many pairs of $\alpha$ and $\beta$ weights to simulate users, ranging from performance-oriented individuals to those with strong cost constraints. The specific weight pairs are presented in Table 9. The small-scale setting involves users 1–9, while the large-scale setting involves 1,000 users (with $\alpha \in \{0.200, 0.201, \ldots, 1.000\}$ and $\beta \in \{0.800, 0.799, \ldots, 0.798\}$), here we only present a subset.

Table 9: **The subset of the weight pairs for simulated users.**

| User | Value of $\alpha$ | Value of $\beta$ |
|---|---|---|
| User 1 | 0.2 | 0.8 |
| User 2 | 0.3 | 0.7 |
| User 3 | 0.4 | 0.6 |
| User 4 | 0.5 | 0.5 |
| User 5 | 0.6 | 0.4 |
| User 6 | 0.7 | 0.3 |
| User 7 | 0.8 | 0.2 |
| User 8 | 0.9 | 0.1 |
| User 9 | 1.0 | 0.0 |

## A.2 Models Used as Judges in Experiments

Under the LLM-as-a-Judge strategy, we employ four LLMs as judges to generate simulated user data. In the small-scale experiments (Section 5.1), we used DeepSeek-V3(DeepSeek-AI et al., 2025) as the LLM judge. In the large-scale experiments (Section 5.2), we used DeepSeek-V3.1, Kimi k2(Team et al., 2025), and Llama-3.3 70B as three LLM judges to reduce potential bias from relying on a single LLM as a judge.

## A.3 Instruction Prompts for LLM to judge the best answer

To instruct the LLM judge to select the response that best aligns with the system prompt, we design two kinds of instruction prompts following (Sun et al., 2025), as shown in Table 10 and Table 11. In the small-scale experiments (Section 5.1), we used the first prompt. In the large-scale experiments (Section 5.2), we used two prompts.

Table 10: **The first instruction prompt for LLM judge with user profile to select the best answer generated by candidate LLMs.**

| **Instruction Prompt for Selection** |
|---|
| Given the Query and m answers, you need to select the best answer that you are most satisfied with. |
| Ensure that the order of the responses does not influence your decision. |
| Do not let the length of the responses impact your evaluation. |
| The system's input is in this format: |
| [User Query] |
| {query} |
| [The Start of Answer 1] |
| {answer_1} |
| [The End of Answer 1] |
| ... |
| [The Start of Answer {m}] |
| {answer_{m}} |
| [The End of Answer {m}] |
| Your response can only include the answer number, ranging from 1 to {m}, no anything else. |

Table 11: **The second instruction prompt for LLM judge with user profile to select the best answer generated by candidate LLMs.**

| **Instruction Prompt for Selection** |
|---|
| You are given a user query and {m} candidate answers. |
| Your task is to carefully read all answers and decide which one best addresses the query in terms of correctness, clarity, and relevance. |
| Important rules: |
| Do not be influenced by the order of the answers. |
| Do not be influenced by the length of the answers. |
| Focus only on the quality and appropriateness of the content. |
| The system input format is: |
| [User Query] |
| {query} |
| [The Start of Answer 1] |
| {answer_1} |
| [The End of Answer 1] |
| ... |
| [The Start of Answer {m}] |
| {answer_{m}} |
| [The End of Answer {m}] |
| Your response must be strictly one integer between 1 and {m}, with no additional text. |

### A.4 User profiles for the Simulated User Profiles

Under the LLM-as-a-Judge strategy, we utilize additional LLM as the judge to simulate diverse user groups with different preferences through user profiles. We used GPT-4o to generate 200 user profiles, the detailed descriptions of which are presented in Table 12. The small-scale setting involves users 1–9, while the large-scale setting involves 200 user prompts; here we only present a subset.

Table 12: **The subset of user profiles for simulated user profiles.**

| User Profile | User Profile |
| --- | --- |
| User 1: Person full of sensibility | You are a person full of sensibility, and you tend to choose answers that are natural, warm, and relatable rather than overly formal or calm expressions. |
| User 2: Inquisitive young person | You are an inquisitive young person, and you prefer answers that are creative and light-hearted with humor. |
| User 3: Math enthusiast | You are a math enthusiast, and you tend to choose answers that are clearly explained, step-by-step, and have a logical process. |
| User 4: Engineer | You are an engineer who prefers answers that are simple and direct, especially those that lead to conclusions through practical calculations and formulae. |
| User 5: Student | You are a student, and you prefer answers that contain detailed explanations and help you understand the concepts. |
| User 6: Information retrieval specialist | You are an information retrieval specialist, and you tend to choose answers that answer the question precisely and where the answer is highly relevant to the context. |
| User 7: News editor | You are a news editor who prefers summaries that contain all the important information, are logical, and are concise. |
| User 8: Literature enthusiast | You are a literature enthusiast who tends to prefer answers that are eloquent, rhetorically rich, and capable of conveying deep emotions. |
| User 9: Expert in childhood education | You are an expert in early childhood education, who prefers explanations that use simple language, are vivid and engaging, easy to understand, and inspiring. |
| User 10: Legal expert | You are a legal expert who values clarity, objectivity, and well-structured arguments. You prefer answers that are precise in language, logically sound, and avoid emotional bias. |
| User 11: Visual artist | You are a visual artist, and you tend to favor answers that use vivid imagery, metaphorical language, and evoke strong sensory impressions. |
| User 12: Executive | You are a busy executive who appreciates answers that are concise, actionable, and get straight to the point without unnecessary elaboration. |
| User 13: Philosopher | You are a philosopher who prefers answers that are thoughtful, nuanced, and show a deep consideration of multiple perspectives. |
| User 14: Healthcare professional | You are a healthcare professional who values answers that are accurate, empathetic, and focused on practical well-being. |
| User 15: Online content creator | You are an online content creator who favors responses that are catchy, emotionally engaging, and easy to share. |

## A.5 Descriptions for Task Datasets and LLMs

To enhance the expressiveness of the initial node embeddings, we use GPT-4o to generate textual descriptions for task datasets and LLMs. These descriptions are then encoded into embedding vectors using a BERT

model, which are used to initialize the GNN. Detailed descriptions are provided in Table 13, Table 14 and Table 15.

Table 13: **The descriptions of task datasets.**

| Task dataset | Description |
| --- | --- |
| Alpaca | The Alpaca dataset is designed for instruction-following tasks, where the model is required to generate coherent and contextually appropriate responses to given instructions or prompts. It focuses on understanding diverse user requests and providing informative and accurate outputs based on those instructions. |
| GSM8K | The GSM8K dataset is tailored for mathematical problem-solving tasks. It consists of natural language math problems that require the model to comprehend the problem statement, apply the correct mathematical operations, and provide the solution. The primary challenge lies in both parsing complex language and performing accurate calculations. |
| SQuAD | The SQuAD dataset is focused on question-answering tasks, where the model is given a passage of text and needs to extract or generate a precise answer to a question based on the content of the passage. The dataset emphasizes comprehension, retrieval of relevant information, and concise answer generation. |
| Multi-News | The Multi-News dataset is aimed at text summarization tasks. It contains multiple news articles on the same topic, and the model's objective is to generate a concise and comprehensive summary that integrates information from all the articles. The challenge is to distill key points while maintaining coherence and avoiding redundancy. |

Table 14: **The descriptions of LLMs (Part 1).**

| LLM | Description |
|---|---|
| LLaMA-3 (8B) | This is a relatively small-sized model (8 billion parameters) designed for general-purpose language tasks. Its low cost per million tokens (0.2) makes it an affordable option for many applications requiring quick responses with moderate accuracy. |
| Mixtral-8x7B | With a combined size of 56 billion parameters, this model aims to provide stronger language modeling capabilities. Its cost per million tokens is 0.6, reflecting its balance between performance and affordability for more complex tasks. |
| NousResearch | A mid-sized model with 34 billion parameters, suitable for handling moderately complex language tasks. Its cost is higher at 0.8 per million tokens, indicating a greater computational demand, likely due to its enhanced capabilities over smaller models. |
| Ministral-8B | A highly efficient model with 8 billion parameters, tailored for fast performance and optimized cost-effectiveness. With a cost of just 0.2 per million tokens, it delivers rapid processing while maintaining exceptional value for resource usage. |
| Mistral-7B | With 7 billion parameters, Mistral-7b is optimized for lightweight tasks, balancing speed and efficiency. Its cost per million tokens is 0.2, making it cost-effective for standard use cases without the need for complex computations. |
| LLaMA-3.1 (8B) | A variant optimized for speed and efficiency with 8 billion parameters. Its cost per million tokens is only 0.2, suggesting that it is designed to handle tasks quickly while being highly cost-effective. |
| LLaMA-3 (70B) | This model, at 70 billion parameters, is tailored for high performance with an emphasis on efficiency. The cost is 0.9 per million tokens, reflecting its advanced capabilities for a broad range of tasks requiring more computation. |
| LLaMA-3.1 (70B) | Large model with 70 billion parameters, likely to offer strong capabilities for various language tasks. Its cost is also 0.9 per million tokens, suggesting similar performance and computational needs as other 70b models. |
| Qwen-2 (72B) | With 72 billion parameters, Qwen-2 is among the largest models in the list, designed for high-complexity tasks. Its cost per million tokens is 0.9, making it comparable to other high-performance models in terms of both capability and expense. |
| Qwen-2.5 7B | Qwen-2.5-7B features 7 billion parameters and is fine-tuned for instruction-following, dialogue, and task completion. It performs well in interactive settings, making it suitable for a wide range of practical applications. |
| Gemma-3 27B | Gemma-3-27B, with 27 billion parameters, is fine-tuned for instruction and dialogue tasks. It combines strong reasoning ability with fluent generation, making it well-suited for advanced interactive applications. |
| Gemma-3 12B | Gemma-3-12B has 12 billion parameters and is optimized for instruction-following and conversational tasks. It offers a balance between capability and efficiency, suitable for a wide range of interactive use cases. |
| LLaMA 4 Scout | LLaMA 4 Scout is a lightweight variant in the LLaMA 4 series, optimized for speed and efficiency. It delivers responsive performance in everyday tasks, making it well-suited for real-time dialogue and low-latency applications |
| phi-4 | Phi-4 is a 14-billion-parameter model focused on high-quality reasoning and language understanding. Built with a compact training dataset, it emphasizes alignment, factuality, and efficient task completion in instruction-driven scenarios. |

Table 15: **The descriptions of LLMs (Part 2).**

| LLM | Description |
| --- | --- |
| Mistral Small 3.2 24B | Mistral Small 3.2 24B is a 24-billion-parameter model built for balanced performance and versatility. It handles a wide range of tasks with strong reasoning and generation capabilities, while maintaining efficiency across general and instruction-based applications. |
| LLaMA-2 (7B) | A compact model at 7 billion parameters, it offers similar capabilities and pricing to LLaMA-3 (7b) at a cost of 0.2 per million tokens. It's an efficient choice for tasks requiring decent performance without high computational costs. |
| LLaMA-3-Turbo (8B) | A variant optimized for speed and efficiency with 8 billion parameters. Its cost per million tokens is only 0.2, suggesting that it is designed to handle tasks quickly while being highly cost-effective. |
| LLaMA-3-Turbo (70B) | This model, at 70 billion parameters, is tailored for high performance with an emphasis on efficiency. The cost is 0.9 per million tokens, reflecting its advanced capabilities for a broad range of tasks requiring more computation. |
| LLaMA-3.1-Turbo (70B) | Large model with 70 billion parameters, likely to offer strong capabilities for various language tasks. Its cost is also 0.9 per million tokens, suggesting similar performance and computational needs as other 70b models. |
| Qwen-1.5 (72B) | With 72 billion parameters, Qwen-1.5 is among the largest models in the list, designed for high-complexity tasks. Its cost per million tokens is 0.9, making it comparable to other high-performance models in terms of both capability and expense. |
| LLaMA-2 (70B) | A larger variant of LLaMA-2, this model has 70 billion parameters, providing advanced capabilities for complex tasks. Its cost per million tokens is 0.9, indicating its higher computational demand and enhanced performance. |
| LLaMA-3.1 (8B) | A variant optimized for speed and efficiency with 8 billion parameters. Its cost per million tokens is only 0.2, suggesting that it is designed to handle tasks quickly while being highly cost-effective. |
| Gemini 2.5 Flash | Gemini 2.5 Flash is a fast and efficient model designed to handle real-time language tasks with quick turnaround times. With 2.5 billion parameters, it delivers solid performance for lightweight applications. |
| Qwen3 Coder 480B A35B | Qwen3 Coder 480B A35B is an extremely large model with 480 billion parameters, engineered for highly specialized coding tasks and advanced problem-solving in programming. |
| GPT-4o-mini | GPT-4o Mini is a compact yet highly capable model designed for a variety of language tasks. With optimized performance in smaller parameter sizes, it offers a strong balance between efficiency and accuracy. |
| Palmyra-Fin | Palmyra-Fin is a domain-specialized language model tailored for financial applications. It is optimized to understand financial terminology, documents, and data, making it particularly effective for tasks like report analysis, market insights, risk evaluation, and financial question answering. |
| Palmyra-Med | Palmyra-Med is a domain-focused model built for the medical and healthcare sector. Trained with extensive biomedical and clinical data, it excels at interpreting medical texts, assisting with clinical decision support, and providing reliable insights in research and healthcare communication. |

## A.6 Candidate LLMs

In our experiments, we accessed the candidate LLMs via the Together AI, OpenRouter AI, and NVIDIA Build. Detailed information is provided in Table 16, Table17 and Table18. For the experiments in Section 5.1 and Section 5.3, we used the candidate LLMs listed in Table 16. For the experiments in Section 5.2 and 5.5, we used the candidate LLMs shown in Table 17. For the experiments in Section 5.4, we used the candidate LLMs shown in Table 18.

It is worth noting that many companies and institutions are rapidly updating their LLMs. As new versions are released, older LLM APIs often become deprecated or unavailable. As a result, we used different sets of candidate LLMs across different experiments. Despite variations in the experimental settings due to changing candidate LLMs, our model consistently demonstrates strong performance, highlighting the generalization capability of `PersonalizedRouter`.

Table 16: **Statistics of candidate LLMs and their costs under two simulation strategies in the experiments of Section 5.1 and Section 5.3.** The LLMs on the left side are called from Together AI, and the LLMs on the right side are called from OpenRouter.

| Scenario | Multi-cost-efficiency Simulation | | Scenario | LLM-as-a-Judge | |
|---|---|---|---|---|---|
| LLM | Size | Cost per 1M tokens | LLM | Size | Cost per 1M tokens |
| LLaMA-3 (7B) | 7B | 0.2 | LLaMA-3 (8B) | 8B | 0.2 |
| Mixtral-8x7B | 56B | 0.8 | Mixtral-8x7B | 56B | 0.6 |
| NousResearch | 34B | 0.6 | NousResearch | 34B | 0.8 |
| LLaMA-2 (7B) | 7B | 0.2 | Mistral-8B | 8B | 0.2 |
| Mistral-7B | 7B | 0.2 | Mistral-7B | 7B | 0.2 |
| LLaMA-3 (70B) | 70B | 0.9 | LLaMA-2 (70B) | 70B | 0.9 |
| LLaMA-3-Turbo (8B) | 8B | 0.2 | LLaMA-3.1 (8B) | 8B | 0.2 |
| LLaMA-3-Turbo (70B) | 70B | 0.9 | LLaMA-3 (70B) | 70B | 0.9 |
| LLaMA-3.1-Turbo (70B) | 70B | 0.9 | LLaMA-3.1 (70B) | 70B | 0.9 |
| Qwen-1.5 (72B) | 72B | 0.9 | Qwen-2 (72B) | 72B | 0.9 |

Table 17: **Statistics of candidate LLMs and their costs under two simulation strategies in the experiments of Section5.2 and Section5.5.** Palmyra-Fin and Palmyra-Med are called via Writer, while the remaining models are accessed through NVIDIA Build.

| LLM | Size | Cost per 1M tokens |
|---|---|---|
| Mixtral-8x7B | 56B | 0.6 |
| Gemini 2.5 Flash | / | 2.0 |
| LLaMA-3.1 (8B) | 8B | 0.2 |
| Qwen3 Coder 480B A35B | 480B | 2.0 |
| Mistral Small 3.2 24B | 24B | 0.4 |
| Palmyra-Fin | 70B | 0.9 |
| Palmyra-Med | 70B | 0.9 |
| Qwen-2.5 7B | 7B | 0.2 |
| Gemma-3 12B | 12B | 0.2 |
| GPT-4o-mini | / | 2.0 |

Table 18: **Statistics of candidate LLMs and their costs under two simulation strategies in the experiments of Section 5.4.** The LLMs are called from OpenRouter.

| LLM | Size | Cost per 1M tokens |
|---|---|---|
| LLaMA-3-8B-instruct | 8B | 0.2 |
| Mixtral-8x7B | 56B | 0.8 |
| Nous-Hermes-2-Mixtral | 34B | 0.6 |
| Mistral-8B | 8B | 0.2 |
| Mistral-7B | 7B | 0.2 |
| LLaMA-3.1-8B-instruct | 8B | 0.2 |
| LLaMA-3-70B-instruct | 70B | 0.9 |
| LLaMA-3.1-70B-instruct | 70B | 0.9 |
| Qwen-2-72B-instruct | 72B | 0.9 |
| Qwen-2.5-7B-instruct | 7B | 0.2 |
| Gemma 3 27B | 27B | 0.6 |
| Gemma 3 12B | 12B | 0.4 |
| LLaMA 4 Scout | 17B | 0.4 |
| phi-4 | 14B | 0.4 |
| Mistral Small 3.2 24B | 24B | 0.4 |

## A.7 The Role of GSM8K and SQuAD in LLM-as-a-Judge Strategy

Since users have distinct preferences over the style of LLM responses, both GSM8K and SQuAD serve as effective benchmarks for evaluating models under the LLM-as-a-Judge setting. We select two sets of cases from GSM8K and SQuAD for discussion, with details presented in Table 19 and Table 20.

In the GSM8K case (shown in Table 19), the response from Qwen-2.5 7B provides a detailed reasoning process but may be redundant for experienced users. In contrast, Gemma-3 12B delivers a concise and logically clear answer, making it more suitable for quick reading.

In the SQuAD case (shown in Table 20), Gemini 2.5 Flash provides the correct answer directly, which is suitable for users who seek quick factual conclusions. In contrast, LLaMA-3.1 (8B) not only provides the correct answer but also supplements it with detailed examples, suiting users who value additional context, such as sports enthusiasts.

Therefore, both GSM8K and SQuAD are useful in evaluating not only in the multi-cost-efficiency simulation strategy but also in the LLM-as-a-Judge strategy.

Table 19: **The comparison of response styles between two LLMs for the same GSM8K query.**

---

**Query:**

Given a simple mathematical question, please directly provide the final answer.

Your response should follow the structure outlined below:

R: <Replace Here With Your Reasonings>;

A: Place your Final Answer here as a clear numeric value. Ensure there are no additional words, signs, or explanations! Enclose the numeric value in angle brackets.

An example of the desired output is:

R: First find the total number of starfish arms: 7 starfish * 5 arms/starfish = «7*5=35»35 arms

Then add the number of seastar arms to find the total number of arms: 35 arms + 14 arms = «35+14=49»49 arms

A: <49>

A small airplane can seat 10 people in first class, 30 in business class, and 50 in economy class seating. If economy class is half full, and business class and first class have the same number of people together as economy class, how many seats are unoccupied in business class if only three people on the flight have first class seats?

---

**Qwen-2.5 7B:**

R: First, we can calculate the number of people in the economy class by multiplying the total number of economy class seats by the fraction of seats filled, which is half (or 0.5):

50 seats * 0.5 = 25 people in economy class.

Given that the number of people in first class and business class combined equals the number of people in economy class, we can say: 25 people = number of people in first class + number of people in business class.

We know that only three people have first class seats, so we can substitute and find the number of business class seats: 25 people = 3 people (first class) + number of people in business class.

From this equation, we can determine the number of people in business class by subtracting the number of first class seats from the total:

number of people in business class = 25 - 3 = 22 people.

We then find the number of unoccupied seats in business class by subtracting the number of people in business class from the total number of seats in business class:

30 seats - 22 people = «30-22=8»8 unoccupied seats.

A: <8>

---

**Gemma-3 12B:**

R: Economy class is half full, so 50 / 2 = 25 people are in economy. Business and first class together have the same number of people as economy, so business and first class have 25 people total. Three people are in first class, so 25 - 3 = 22 people are in business class. Business class has 30 seats, so 30 - 22 = 8 seats are unoccupied.

A: <8>

---

Table 20: **The comparison of response styles between two LLMs for the same SQuAD query.**

| **Query:** |
| --- |
| How many appearances have the Denver Broncos made in the Super Bowl? |
| **Gemini 2.5 Flash:** |
| The Denver Broncos have made 8 appearances in the Super Bowl. |
| **LLaMA-3.1 (8B):** |
| The Denver Broncos have made 8 appearances in the Super Bowl: |
| |
| 1. 1967: Lost to the Green Bay Packers 33-14 in Super Bowl II |
| 2. 1977: Lost to the Dallas Cowboys 27-10 in Super Bowl XII |
| 3. 1987: Lost to the New York Giants 39-20 in Super Bowl XXI |
| 4. 1988: Lost to the Washington Redskins 42-10 in Super Bowl XXII |
| 5. 1989: Lost to the San Francisco 49ers 55-10 in Super Bowl XXIV |
| 6. 1997: Lost to the Green Bay Packers 35-21 in Super Bowl XXXII |
| 7. 1998: Defeated the Atlanta Falcons 34-19 in Super Bowl XXXIII and won their first Super Bowl title |
| 8. 2015: Lost to the Seattle Seahawks 43-8 in Super Bowl XLVIII |

### A.8 T-SNE Visualization of Routing Decisions

To further analyze the explainability of our routing mechanism, we evaluate it under the multi-cost-efficiency simulation strategy. We consider a single query and examine routing outcomes across three candidate LLMs for ten users. Among these, three users adopt cost-oriented preferences ($\alpha = 0.21, 0.22, 0.23$), while the remaining seven favor performance-oriented preferences ($\alpha = 0.94, 0.95, \ldots, 1.00$).

We leverage the LLM embeddings $\mathbf{h}_m$ alongside the combined user–query–task embeddings $\mathbf{h}_{uqt}$ (introduced in Section 3.2). Both sets of embeddings are reduced to two dimensions via t-SNE and jointly visualized within the same coordinate system (Figure 3). This setup highlights how the router allocates different LLMs for the same query depending on user preference.

The results show that users 0–2 (cost-oriented) are near LLM2, whereas the remaining performance-oriented users are routed to LLM0 and LLM1. Importantly, users with strongly divergent preferences are well separated in the embedding space and assigned to different models. This confirms that `PersonalizedRouter` can successfully infer latent user preferences and enable personalized routing.

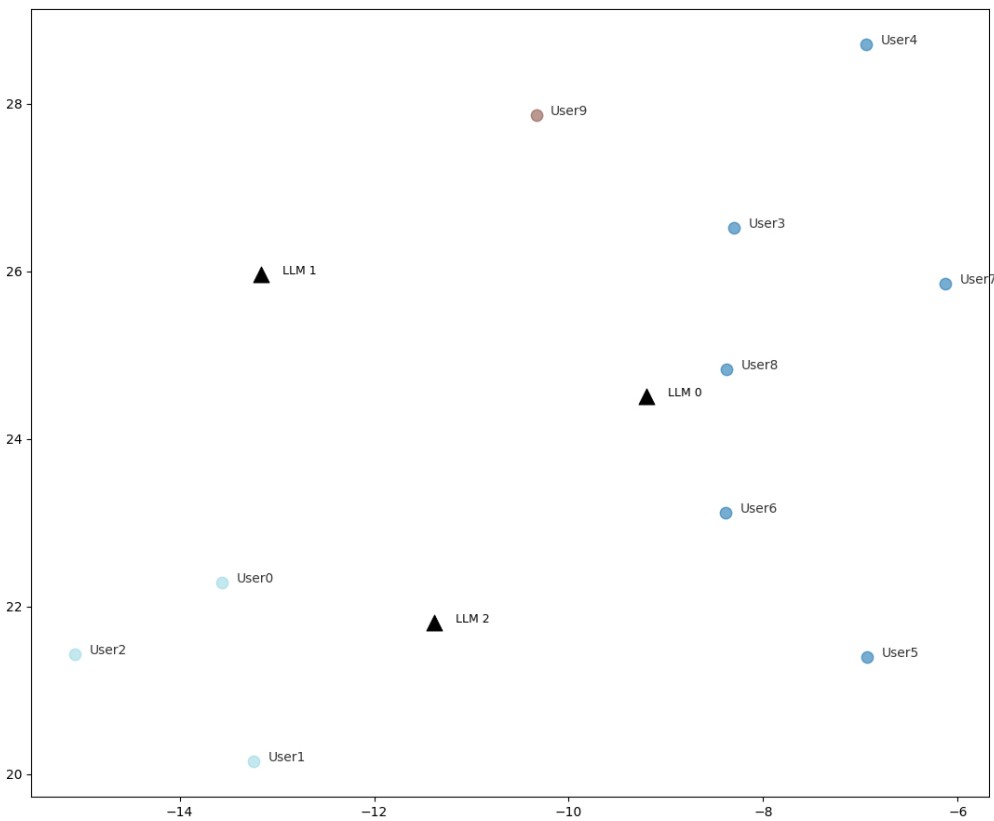

Figure 3: **T-SNE Visualization of Routing Decisions**. In the visualization, user embeddings are represented by circles, LLM embeddings by triangles, and the color indicates the assignment to a particular LLM. Users 0–2 are cost-oriented, while users 3–9 are performance-oriented. `PersonalizedRouter` successfully learns the latent preferences of users, separates the two groups, and ultimately performs personalized routing.

## A.9  Real User Questionnaire Template

To mitigate the potential bias of simulated users, we collected a small-scale real-user interaction dataset consisting of 40 participants and 80 queries, which was used to validate the generalization and feasibility of `PersonalizedRouter` in real-world user scenarios. The questionnaire template used for data collection is shown in Table 21, where each page contains one query and the corresponding ten responses generated by different LLMs. Participants were required to fill in the ID of their preferred response at the bottom of the page. Due to space limitations, only four answers are shown in the questionnaire template.

Table 21: **Template of User Preference Questionnaire**. For each query, participants are asked to choose their most preferred answer from the 10 candidates.

---

**Query:** What are the three primary colors?

**Answer 0:**
The three primary colors are:

1. **Red**
2. **Blue**
3. **Yellow**
These colors cannot be created by mixing other colors together, and they are the base colors used to create all other colors. By mixing different combinations of primary colors, you can produce a wide range of secondary colors and tertiary colors.

**Answer 1:**
The three primary colors in additive color theory (used in light, like computer screens) are red, green, and blue. When combined in varying intensities, they can create virtually any color perception.

However, it's important to note that in subtractive color theory (used in paints and pigments), the primary colors are typically considered to be cyan, magenta, and yellow. When combined, these subtractive primary colors can create a wide spectrum of colors as well.

So, the primary colors can be red, green, and blue or cyan, magenta, and yellow, depending on whether you're dealing with additive or subtractive color systems.

**Answer 2:**
The three primary colors are red, blue, and yellow.

**Answer 3:**
The three primary colors are:

- Red
- Blue
- Yellow
These are the colors that cannot be created by mixing other colors. All other colors can be created by mixing these three primary colors in various combinations.

**Answer 4:**
The three primary colors are:

1. **Red**
2. **Blue**
3. **Yellow**
These colors are called primary because they cannot be created by mixing other colors together. They are the base colors that all other colors are derived from.
Do you have any other questions about colors?

**Answer 5:** [Text generated by LLM 5]
**Answer 6:** [Text generated by LLM 6]
**Answer 7:** [Text generated by LLM 7]
**Answer 8:** [Text generated by LLM 8]
**Answer 9:** [Text generated by LLM 9]
**Your Favorite Answer:**

---

