# OpenReview forum: "PersonalizedRouter: Personalized LLM Routing via Graph-based User Preference Modeling"
_TMLR — Accepted by TMLR_

### Review · Reviewer_h7mj · 2025-08-20

**Summary Of Contributions:**

This paper proposes a new method to assign a "good" LLM to a specific user by accounting for individual per-task LLM performance and costs.

**strengths**
- clearly defined problem setup
- well motivated graph construction

**weaknesses**
- Connections with GraphRouter are not well discussed: This work seems to use an identical (or near identical) network and graph setup compared to GraphRouter, but does not disclose this. Further, GraphRouter is not used as a baseline in the experiments.
- This paper claims "[Current routers] fail to learn user preferences from interaction data" in the abstract: However, if this is a core motivation, then a preference based method such as RouteLLM would be more appropriate
- The experiments are really small scale:  Just using 10 LLMs may arguably be fine, but just 9 users is too small by many orders of magnitude. In reality one might have thousands of users to route between the 10 LLMs (in the open router setting you might even have millions), while I do understand that training in the million user regime is intractable, one should still look at at least 1000 users (probably more) to showcase the router can potentially scale to such instances. Looking at the work right now, my fear is that the setup essentially designs dense tripartite graphs which might quickly reach their scaling limits.
- The LLM-as-a-Judge method may introduce arbitrary bias into model evaluation due to the prompt design and the LLM judge model used. Arguably the "correct" way to use such a judge might be to check multiple LLMs and multiple prompts and combine their rankings.
- The models are all "mid tier" models without any frontier LLMs from e.g. anthropic, openai, qwen or deepseek. This means that the range of performance is not as wide as it could be making assignment easier than what it should be. The LLMs are also all generalist models, rather than having e.g. experts in different domains (e.g. via low-rank adaptation) that might give more depth to the routing (e.g. use a small, cheap domain specific model rather than a generalist, but expensive frontier LLM)

**Audience:**

Yes

**Audience Explanation:**

I do think that having a capable router is an interesting research direction

**Broader Impact Concerns:**

We have no broader impact concerns.

**Claims And Evidence:**

No

**Claims Explanation:**

The scale of the LLM evaluation is insufficiently small, meaning that one cannot judge the performance at the scale necessary for real systems.

**Requested Changes:**

- Larger scale experiments (more users)
- Make connections with GraphRouter clear, also include in benchmarks
- discuss why direct preference methods (such as RouteLLM) "fail to learn user preferences from interaction data"
- Increase the heterogeneity of LLMs (include frontier models, domain-expert models, etc...)

---

> ### Author Response · Authors · 2025-09-19
> **Response to Reviewer h7mj (Part 1)**
>
> **Q1. Connections with GraphRouter are not well discussed: This work seems to use an identical (or near identical) network and graph setup compared to GraphRouter, but does not disclose this. Further, GraphRouter is not used as a baseline in the experiments.**
>
> **Response:** Thanks for your valuable questions and insightful feedback. We answer your questions in two parts.
>
> **[Connections with GraphRouter]:** PersonalizedRouter and GraphRouter differ fundamentally in their application scenarios. GraphRouter performs LLM routing using fixed cost/performance weight pairs (details are provided in Section 4.2), which restricts it to static user settings. As stated in our paper (Section 1, Instruction, Paragraph 2):
>
> >“GraphRouter constructs a heterogeneous graph from user interaction data and leverages a GNN to predict the most suitable LLM. However, a fundamental limitation is its inability to adequately model user preferences (see Table 1).”
>
> When extended to multi-user scenarios, GraphRouter would require training a separate model for each individual user, which is computationally inefficient and impractical. In contrast, PersonalizedRouter is explicitly designed for multi-user and personalized settings. It introduces user nodes into the graph (details are provided in Section 3.2), and edges are used to represent different types of preferences (e.g., performance-oriented, style-oriented). This enables PersonalizedRouter to learn latent user preferences from interaction data and achieve personalized LLM selection across multiple users, which is not considered in prior work, to the best of our knowledge.
>
> **[GraphRouter is not used as a baseline in the experiments.]:** In the revised version of the paper, we have also included GraphRouter as a baseline and conducted additional experiments under both the multi-cost-efficiency simulation strategy and the LLM-as-a-Judge strategy.
>
> **Q2. a. This paper claims "[Current routers] fail to learn user preferences from interaction data" in the abstract: However, if this is a core motivation, then a preference based method such as RouteLLM would be more appropriate.**
>
> **b. discuss why direct preference methods (such as RouteLLM) "fail to learn user preferences from interaction data"**
>
> **Response:** We thank the reviewer for pointing this out and have revised the statement in the latest version to:
>
> >“[Current routers] fail to learn individual user preferences from interaction data.”
>
>  It is important to note that our motivation is to enable LLM selection in multi-user routing scenarios by learning each user’s preferences over cost/performance trade-offs or response styles. RouteLLM is trained on Chatbot Arena preference data, which consists of votes from all users, but it does not model individual preferences at inference time. Moreover, RouteLLM’s inference relies on the performance gap between a strong and a weak model together with the cost threshold α, which corresponds to the first part of the original sentence.
>
> >“Current LLM selection methods typically optimize for a single fixed objective, such as performance, cost, or a trade-off between them, and fail to learn user preferences from interaction data.”
>
>  In addition, RouteLLM is only applicable to binary routing between one strong and one weak LLM, and therefore cannot address multi-user, multi-LLM routing scenarios.
>
> **Q3. a. The experiments are really small scale: Just using 10 LLMs may arguably be fine, but just 9 users is too small by many orders of magnitude. In reality one might have thousands of users to route between the 10 LLMs (in the open router setting you might even have millions), while I do understand that training in the million user regime is intractable, one should still look at at least 1000 users (probably more) to showcase the router can potentially scale to such instances. Looking at the work right now, my fear is that the setup essentially designs dense tripartite graphs which might quickly reach their scaling limits.**
>
> **b. The LLM-as-a-Judge method may introduce arbitrary bias into model evaluation due to the prompt design and the LLM judge model used. Arguably the "correct" way to use such a judge might be to check multiple LLMs and multiple prompts and combine their rankings.**
>
> **c. Larger scale experiments (more users)**
>
> **Response:** Thanks for raising the above concerns. Following your suggestion, we conducted new large-scale experiments under both the multi-cost-efficiency simulation strategy and the LLM-as-a-Judge strategy. Under the multi-cost-efficiency simulation strategy, we evaluated 10 LLMs across 1,000 users. We also carried out new LLM-as-a-Judge experiments. To further mitigate the bias of relying on a single LLM as judge, we employed 3 different LLMs, each combined with two types of prompts, yielding 6 judge settings. The experimental results are presented in Table1.

---

> ### Author Response · Authors · 2025-09-19
> **Response to Reviewer h7mj (Part 2)**
>
> **Table1: Comparison between different methods under two simulation strategies in the large-scale experimental setting. The Deltas represent the improvement of the Reward over the best baseline. The Reduction indicates the relative reduction in time cost compared to the baseline with the highest time cost.**
>
> | Scenario    | multi-cost-efficiency simulation             | LLM-as-a-Judge strategy                |
> |-------------|----------------------------------------------|----------------------------------------|
> | Method      | Reward &nbsp;  Delta (%) &nbsp;  Time   &nbsp;  Reduction (%)   | Accuracy &nbsp;  Delta (%) &nbsp;  Time   &nbsp; Reduction (%) |
> | HybridLLM   | 0.119  &emsp;  -43.33    &ensp;&emsp;  4:32   &emsp;&emsp;  98.50  &nbsp;         | 0.196    &emsp;&emsp;&emsp;  0.00      &emsp;&emsp;  4:56   &emsp;&nbsp; 98.40         |
> | FrugalGPT   | 0.141  &emsp;  -32.86    &ensp;&emsp;  47:44  &emsp;&emsp;  84.29  &nbsp;         | 0.108    &emsp;&emsp; -44.90     &emsp;&emsp;  49:31  &ensp;&nbsp; 83.92         |
> | GraphRouter | 0.204  &emsp;  -2.86     &ensp;&emsp;&ensp;  8:26   &emsp;&emsp;  97.22  &nbsp;         | 0.137    &emsp;&emsp; -30.10     &emsp;&emsp;  9:28   &emsp;&ensp; 96.93         |
> |RouterDC | 0.210  &nbsp;&emsp; 0.00     &ensp;&emsp;  303:47 &emsp;&emsp;  0.00   &nbsp;         | 0.153    &emsp;&emsp; -21.94     &emsp;&emsp;  308:02 &ensp;&nbsp; 0.00          |
> | Ours        | 0.244  &emsp;   16.19    &ensp;&emsp; 10:15   &emsp;&emsp; 96.63   &nbsp;         | 0.313    &emsp;&emsp;&nbsp;  59.69     &emsp;&emsp;  11:37  &emsp;&nbsp; 96.23         |
> | Oracle      | 0.310  &emsp;   47.62    &ensp;&emsp;  /      &emsp;&emsp;  /      &nbsp;         | 1.000    &emsp;&emsp; 410.20     &emsp;&emsp;  /      &ensp;&nbsp; /             |
>
> **Q4. a.The models are all "mid tier" models without any frontier LLMs from e.g. anthropic, openai, qwen or deepseek. This means that the range of performance is not as wide as it could be making assignment easier than what it should be. The LLMs are also all generalist models, rather than having e.g. experts in different domains (e.g. via low-rank adaptation) that might give more depth to the routing (e.g. use a small, cheap domain specific model rather than a generalist, but expensive frontier LLM)**
>
> **b. Increase the heterogeneity of LLMs (include frontier models, domain-expert models, etc...)**
>
> **Response:**  Thank you for your careful review. In the new large-scale experiments, we updated the set of candidate models. For frontier models, we introduced Gemini 2.5 Flash, GPT-4o-mini, and Qwen3 Coder 480B A35B. For domain-specific expert models, we added Qwen3 Coder 480B A35B, palmyra-fin, and palmyra-med. As a result, our new experiment covers 10 LLMs: Mixtral-8x7B, Gemini 2.5 Flash, LLaMA-3.1 (8B), Qwen3 Coder 480B A35B, Mistral Small 3.2 24B, palmyra-fin, palmyra-med, Qwen-2.5 7B, Gemma-3 12B, and GPT-4o-mini. Information about the candidate LLMs is provided in Appendix A.6 of the latest revision.
>
> **Q5. Make connections with GraphRouter clear, also include in benchmarks.**
>
> **Response:**  Thanks for your constructive feedback. As clarified in response to Q1, PersonalizedRouter learns latent user preferences from interaction data, enabling personalized LLM selection across multiple users. In contrast, GraphRouter is restricted to fixed-user routing. Furthermore, PersonalizedRouter augments the graph with user nodes, and edges encode different types of preferences (e.g., performance-oriented, style-oriented).
>
> In our large-scale experiments under both the multi-cost-efficiency simulation strategy and the LLM-as-a-Judge strategy, we have included GraphRouter as part of the benchmark for comparison. The experimental results of large-scale experiments are shown in Table 1. We also added GraphRouter experiments under other settings, with detailed results reported in Tables 2,3,4, and 5 below. Our PersonalizedRouter still outperforms all other baselines.
>
> **Table2:Comparison between different methods under small-scale human interaction dataset. The Improvement is measured relative to the best baseline.**
>
> | Scenario    |       LLM-as-a-Judge strategy      |
> |-------------|------------------------------------|
> | Method      |       Accuracy &nbsp;  Improvement (%)   |
> | HybridLLM   |       0.315  &emsp;&emsp;&ensp;   -23.73             |
> | FrugalGPT   |       0.350  &emsp;&emsp;&ensp;   -15.25             |
> | GraphRouter |       0.357  &emsp;&emsp;&ensp;   -13.56             |
> | RouterDC    |       0.413  &emsp;&emsp;&ensp;&nbsp;    0.00              |
> | PersonalizedRouter| 0.438  &emsp;&emsp;&ensp;&nbsp;    6.05              |
> | Oracle      |       1.000  &emsp;&emsp;&ensp;   142.13             |

---

> ### Author Response · Authors · 2025-09-19
> **Response to Reviewer h7mj (Part 3)**
>
> **Table3: Comparison between different methods under two simulation strategies in the small-scale experimental setting. The Improvement is measured relative to the best baseline.**
>
> | Scenario&nbsp; | Multi-cost-efficiency Simulation      | LLM-as-a-Judge    |
> |----------------|---------------------------------------|-------------------|
> | Method     | Reward &nbsp;  Improvement (%) | Accuracy &nbsp;  Improvement (%)  |
> | HybridLLM  | 0.141  &emsp;&emsp;&emsp; -36.20           | 0.347  &emsp;&emsp;&emsp;   -14.74            |
> | FrugalGPT  | 0.218  &emsp;&emsp;&emsp; -1.36            | 0.354  &emsp;&emsp;&emsp;   -13.22            |
> | GraphRouter| 0.221  &emsp;&emsp;&emsp; 0.00             | 0.364  &emsp;&emsp;&emsp;   -10.57            |
> | RouterDC   | 0.208  &emsp;&emsp;&emsp; -5.88            | 0.407  &emsp;&emsp;&emsp;   0.00              |
> | Ours       | 0.255  &emsp;&emsp;&emsp; 15.38            | 0.447  &emsp;&emsp;&emsp;   9.83              |
> | Oracle     | 0.304  &emsp;&emsp;&emsp; 37.56            | 1.000  &emsp;&emsp;&emsp;   145.70            |
>
>
> **Table4: Comparison between different methods under simulation strategies in the new user experimental setting. The Improvement is measured relative to the best baseline.**
>
> | Scenario         | multi-cost-efficiency simulation             | LLM-as-a-Judge strategy                |
> |-------------     |----------------------------------------------|----------------------------------------|
> | Method           | Reward  &nbsp; Improvement (%) | Accuracy   Improvement (%)  |
> | HybridLLM        | -0.142   &emsp;&emsp;&emsp;   -240.59         | 0.294  &emsp;&emsp;&emsp;&ensp;    0.00             |
> | FrugalGPT        | 0.044    &emsp;&emsp;&emsp;   -56.44          | 0.192  &emsp;&emsp;&emsp;   -34.69            |
> | GraphRouter      | 0.083    &emsp;&emsp;&emsp;   -17.82          | 0.256  &emsp;&emsp;&emsp;   -12.93            |
> | RouterDC         | 0.101    &emsp;&emsp;&emsp;&ensp;    0.00           | 0.208  &emsp;&emsp;&emsp;   -29.25            |
> | Ours (Few-shots) | 0.070    &emsp;&emsp;&emsp;  -30.69           | 0.313  &emsp;&emsp;&emsp;&ensp;    6.46             |
> | Ours (Trained)   | 0.108    &emsp;&emsp;&emsp;&ensp;   6.93            | 0.326  &emsp;&emsp;&emsp;&ensp;   10.88             |
> | Oracle           | 0.116    &emsp;&emsp;&emsp;&ensp;   14.85           | 1.000  &emsp;&emsp;&emsp;&ensp;   240.14
>
> **Table5: Comparison between different methods under simulation strategies in the new LLM experimental setting. The Improvement is measured relative to the best baseline.**
>
> | Scenario        | multi-cost-efficiency simulation             | LLM-as-a-Judge strategy                |
> |-------------    |----------------------------------------------|----------------------------------------|
> | Method          | Reward &nbsp;  Improvement (%) | Accuracy &nbsp;  Improvement (%)  |
> | HybridLLM       | 0.137 &emsp;&emsp;&emsp;  -37.16           | 0.273  &emsp;&emsp;&emsp;    0.00             |
> | FrugalGPT       | 0.182 &emsp;&emsp;&emsp;  -16.51           | 0.038  &emsp;&emsp;&emsp;   -86.08            |
> | GraphRouter     | 0.203 &emsp;&emsp;&emsp;  -6.88            | 0.247  &emsp;&emsp;&emsp;   -9.52             |
> | RouterDC        | 0.218 &emsp;&emsp;&emsp;   0.00            | 0.199  &emsp;&emsp;&emsp;  -27.10             |
> | Ours (few-shot) | 0.201 &emsp;&emsp;&emsp;  -7.80            | 0.329  &emsp;&emsp;&emsp;   20.51             |
> | Ours (trained)  | 0.234 &emsp;&emsp;&emsp;   7.34            | 0.383  &emsp;&emsp;&emsp;   40.29             |
> | Oracle          | 0.371 &emsp;&emsp;&emsp;  70.18            | 1.000  &emsp;&emsp;&emsp;  266.30             |

---

> > ### Comment · Reviewer_h7mj · 2025-10-01
> >
> > Thank you for your very thorough additional experiments. I think you were able to address most of my concerns. I just have two points to make:
> > Regarding the preference based comparison: My argument regarding RouteLLM was less that this specific model would be used, but rather that the mathematical model for preference that they assume seems to be more fitting than the one proposed here. RouteLLM uses the framework of preference modelling to produce the distribution over "winning" LLMs in (in their case) a price vs performance setting. This makes sense due to the decision theoretic underpinnings of preference optimization (indeed maximizing a utility and encoding preferences is equivalent via the von Neumann–Morgenstern utility theorem). RouteLLM's modelling essentially introduces a utility function (in their case the log-likelihood), which implicitly encodes a preference for one vs the other model. The reason they do not capture user preferences is because they do not condition their distribution on a "user embedding" (a very concrete example of this happening is in RLHF reward models which optimizes e.g. a bradley-terry model over the preferences. Route LLM doesn't need this since it has ground-truth binary preferences, but from a decision theory point of view this is identical, you just assume absolute rather than relative feedback).
> >
> > I'm unsure whether a link-prediction task is really ideal for the problem as described: The method used by e.g. RouteLLM also introduces a preference graph, but in an implicit fashion. However, I do want to stress that I don't see this as a blocker for this paper's acceptance: The fact that the link-prediction problem might not be ideal does not mean this paper is unpublishable!
> >
> > The second remark is regarding the "1000 User setting": How did you define the users in that setting? I see that you have a table in the appendix for 15 users, but how are you generating the personas for the 1000 user setting?
> >
> > Other than that, your updates seem to have covered my concerns.

---

> ### Author Response · Authors · 2025-10-02
> **Response to Preference Modeling and User Simulation**
>
> Thanks for your valuable feedback. We have uploaded the latest revised version.
>
> **Q1.**
> We agree that RouteLLM introduces a theoretically grounded utility function to make routing decisions between two candidate LLMs. However, in real-world applications, it is often necessary to handle routing among multiple candidate LLMs simultaneously. (e.g., AI coding tools that integrate several LLMs). Moreover, [1] and [2] point out that restricting routing methods to a limited pool of models is a limitation. RouteLLM is restricted to binary routing and cannot naturally extend to multi-LLM routing. To address this issue, we follow the GraphRouter and further explore a graph-based approach. PersonalizedRouter mainly focuses on multi-user and multi-LLM settings, leveraging user interaction information to capture latent user preferences for personalized routing. In contrast, RouteLLM treats all users identically in its routing process, restricting personalization in settings with multiple users. Nevertheless, we highly appreciate the reviewer’s insightful comments, and in the revised version we have added a discussion in the Limitation section about the utility function.
>
> **[1]** Feng, T., Shen, Y., & You, J. GraphRouter: A Graph-based Router for LLM Selections. In The Thirteenth International Conference on Learning Representations.
>
> **[2]** Tran, C., Paracha, S., Hafeez, A., & Chen, S. (2025). Arch-Router: Aligning LLM Routing with Human Preferences. arXiv preprint arXiv:2506.16655.
>
>
> **Q2.**
> **In the multi-cost-efficiency simulation strategy,** each weight pair represents one user. We set 1,000 different weight pairs to capture diverse user preferences over performance and cost. The details are presented in Appendix A.1. **In the LLM-as-a-Judge strategy,** the output generated under a given judge configuration for a user profile corresponds to one user. The details are provided in Section 5.2. We selected three different LLMs and applied two distinct instruction prompts to each model, resulting in six judge configurations. For each configuration, we simulated users based on 200 user profiles, yielding a total of 1,200 simulated user preferences across the six configurations. **In other words, each user profile is evaluated under six distinct judge configurations, effectively yielding six different user perspectives.** Ultimately, the combination of 200 user profiles with six judge configurations results in a total of 1,200 simulated user samples.
> We used GPT-4o to generate 200 user profiles. Since presenting all 200 user profiles would be excessive, we only show 15 examples in Table 12 of the Appendix. In the latest revised version, we have supplemented Appendix A.4 with the method for generating the user profiles.

---

### Review · Reviewer_iphw · 2025-08-27

**Summary Of Contributions:**

The paper proposes a graph-based framework for routing queries to the most suitable LLM by modeling user preferences. It builds a heterogeneous graph from user  LLM interactions and applies a GNN to infer hidden user preferences. Two evaluation strategies are introduced: multi-cost-efficiency (balancing accuracy vs. cost) and LLM as a Judge (capturing style preferences). Experiments show it outperforms baselines by up to 16.97% and generalizes well to new users (64.8%) and new LLMs (85.9%). Overall, it enables scalable, personalized, and efficient LLM selection for diverse user needs.

**Audience:**

Yes

**Audience Explanation:**

TMLR audiences would be interested in this work because it addresses a highly relevant challenge: selecting the right LLM for diverse users in an era where models are rapidly proliferating. The paper introduces a personalization framework using graph-based modeling to capture user preferences. The results are compelling, showing significant accuracy improvements (up to 16.97%) and efficiency gains (up to 96%), while also demonstrating few-shot adaptability to new users and unseen LLMs.

**Broader Impact Concerns:**

As briefly mentioned in limitations section, personalized LLM routing risks reinforcing user biases by overfitting to narrow preferences. It may also capture sensitive attributes from interaction data, raising privacy and ethical concerns. Optimizing for efficiency could trade off against fairness and safety in high-stakes domains.

**Claims And Evidence:**

Yes

**Claims Explanation:**

The paper benchmarks PersonalizedRouter against strong baselines and shows consistent improvements, e.g., 16.97% higher reward and 9.83% higher accuracy. Results are compared to an oracle upper bound. Scalability tests with more users and LLMs show that the model maintains high accuracy while cutting time cost. Generalization experiments prove robustness, with few-shot adaptation reaching 64.8% (new users) and 85.9% (new LLMs) of full performance.

**Requested Changes:**

Here some suggestions for the authors:
1. Add a small human user study to validate simulated preferences.
2. Extend the model to handle dynamic, evolving user preferences.
3. Improve explainability of routing decisions (e.g., why specific LLM is chosen).

---

> ### Author Response · Authors · 2025-09-19
> **Response to Reviewer iphw (Part 1)**
>
> **Q1. Add a small human user study to validate simulated preferences.**
>
> **Response:**  Thanks for your constructive feedback. Following your constructive suggestion, we additionally collected a small-scale real-user dataset consisting of 10 LLMs, 40 users, and 80 queries, and conducted an additional LLM-as-a-Judge experiment. The detailed results are presented in Table 1. These results demonstrate that PersonalizedRouter generalizes beyond simulated settings and is also effective in real-user scenarios.
>
> **Table1:Comparison between different methods under small-scale human interaction dataset. The Improvement is measured relative to the best baseline.**
>
> | Scenario    |       LLM-as-a-Judge strategy      |
> |-------------|------------------------------------|
> | Method      |       Accuracy &nbsp;  Improvement (%)   |
> | HybridLLM   |       0.315  &emsp;&emsp;&ensp;   -23.73             |
> | FrugalGPT   |       0.350  &emsp;&emsp;&ensp;   -15.25             |
> | GraphRouter |       0.357  &emsp;&emsp;&ensp;   -13.56             |
> | RouterDC    |       0.413  &emsp;&emsp;&ensp;&nbsp;    0.00              |
> | PersonalizedRouter| 0.438  &emsp;&emsp;&ensp;&nbsp;    6.05              |
> | Oracle      |       1.000  &emsp;&emsp;&ensp;   142.13             |
>
> **Q2. Extend the model to handle dynamic, evolving user preferences.**
>
> **Response:** Thanks for your constructive advice. Real users may make different choices for the same query due to dynamic psychological states. To better simulate this realistic setting, we conducted additional LLM-as-a-Judge experiments. In the new experimental setting, we employed three LLMs, each evaluated with two types of instruction prompts. Consequently, each user query was evaluated under six different LLM-judge settings, thereby simulating the variability of real users’ psychological states when responding to the same query. This design allows us to assess whether our current model can effectively handle dynamic user preferences. The results of these new LLM-as-a-Judge experiments are shown in Table2 , demonstrating that the current model performs well in the dynamic user preference experimental setting.
>
> **Table2: Comparison between different methods under two simulation strategies in the large-scale experimental setting. The Deltas represent the improvement of the Reward over the best baseline. The Reduction indicates the relative reduction in time cost compared to the baseline with the highest time cost.**
>
> | Scenario    | multi-cost-efficiency simulation             | LLM-as-a-Judge strategy                |
> |-------------|----------------------------------------------|----------------------------------------|
> | Method      | Reward &nbsp;  Delta (%) &nbsp;  Time   &nbsp;  Reduction (%)   | Accuracy &nbsp;  Delta (%) &nbsp;  Time   &nbsp; Reduction (%) |
> | HybridLLM   | 0.119  &emsp;  -43.33    &ensp;&emsp;  4:32   &emsp;&emsp;  98.50  &nbsp;         | 0.196    &emsp;&emsp;&emsp;  0.00      &emsp;&emsp;  4:56   &emsp;&nbsp; 98.40         |
> | FrugalGPT   | 0.141  &emsp;  -32.86    &ensp;&emsp;  47:44  &emsp;&emsp;  84.29  &nbsp;         | 0.108    &emsp;&emsp; -44.90     &emsp;&emsp;  49:31  &ensp;&nbsp; 83.92         |
> | GraphRouter | 0.204  &emsp;  -2.86     &ensp;&emsp;&ensp;  8:26   &emsp;&emsp;  97.22  &nbsp;         | 0.137    &emsp;&emsp; -30.10     &emsp;&emsp;  9:28   &emsp;&ensp; 96.93         |
> |RouterDC | 0.210  &nbsp;&emsp; 0.00     &ensp;&emsp;  303:47 &emsp;&emsp;  0.00   &nbsp;         | 0.153    &emsp;&emsp; -21.94     &emsp;&emsp;  308:02 &ensp;&nbsp; 0.00          |
> | Ours        | 0.244  &emsp;   16.19    &ensp;&emsp; 10:15   &emsp;&emsp; 96.63   &nbsp;         | 0.313    &emsp;&emsp;&nbsp;  59.69     &emsp;&emsp;  11:37  &emsp;&nbsp; 96.23         |
> | Oracle      | 0.310  &emsp;   47.62    &ensp;&emsp;  /      &emsp;&emsp;  /      &nbsp;         | 1.000    &emsp;&emsp; 410.20     &emsp;&emsp;  /      &ensp;&nbsp; /             |

---

> > ### Author Response · Authors · 2025-09-19
> > **Response to Reviewer iphw (Part 2)**
> >
> > **Q3. Improve explainability of routing decisions (e.g., why specific LLM is chosen).**
> >
> > **Response:**
> > To better illustrate the explainability of our routing mechanism, we conducted an analysis under the multi-cost-efficiency simulation strategy. For a given query, we analyzed the routing decisions among three candidate LLMs across ten users. Among these users, three exhibited a cost-prioritized preference ($\alpha$=0.21,0.22,0.23), while the other seven showed a performance-prioritized preference ($\alpha$=0.94,0.95,…,1.00).
> >
> > We used the LLM embeddings $h_m$ and the embeddings of user, query, and task $h_{uqt}$ (as introduced in Section 3.2)​. By applying t-SNE, we project $h_{uqt}$ and $h_m$ from the high-dimensional space into two dimensions and visualize them in the same coordinate system. We aim to explain how the router assigns different LLMs to the same query depending on user preferences. The visualization results are provided in Appendix A.8 of the revised paper, where circular markers denote users and triangular markers denote LLMs. Users assigned to the same LLM are indicated with the same color.
> >
> > As shown in the visualization, users 0–2 (cost-prioritized) were assigned to LLM2, as their embeddings $h_{uqt}$​ were closer to the embedding $h_m$​ of LLM2. Similarly, user 9 was assigned to LLM1, while the remaining performance-prioritized users were distributed across LLM0. Notably, users with highly divergent preferences are clearly separated and allocated to different LLMs, demonstrating that PersonalizedRouter can effectively learn latent user preferences and perform personalized routing.
> >
> > **Q4. As briefly mentioned in limitations section, personalized LLM routing risks reinforcing user biases by overfitting to narrow preferences. It may also capture sensitive attributes from interaction data, raising privacy and ethical concerns. Optimizing for efficiency could trade off against fairness and safety in high-stakes domains.**
> >
> > **Response:**   Thanks for the reviewer’s insightful question. Personalized LLMs often face such challenges, which remain an open question and are difficult to address perfectly. Our work primarily focuses on tackling personalized LLM routing in multi-user scenarios. Within this scope, we have made the following efforts, and we will answer your questions one by one.
> >
> > **[ personalized LLM routing risks reinforcing user biases by overfitting to narrow preferences.]:**  We clarify that in both simulation settings, user preferences are expressed through either multiple weight pairs or diverse persona prompts, which encourages the model to learn diverse user preferences. Experimental results in the new-user setting also show that PersonalizedRouter generalizes well to unseen users, rather than being confined to a single user preference.
> >
> > **[It may also capture sensitive attributes from interaction data, raising privacy and ethical concerns.]:**  The model learns preferences from query–LLM interaction records, enabling more accurate prediction of the most suitable LLM for a given query. We will leave addressing the challenge of accessing private data to future work.
> >
> > **[Optimizing for efficiency could trade off against fairness and safety in high-stakes domains.]:**  Our findings show that PersonalizedRouter achieves strong performance in low-risk domains. In future work, we will try to extend it to high-risk applications, with a particular emphasis on fairness and safety.

---

### Review · Reviewer_2sp7 · 2025-09-03

**Summary Of Contributions:**

The paper introduces PersonalizedRouter, a graph-based framework for personalized LLM routing. Its contributions are:
1. Formulating the LLM selection problem in multi-user scenarios, where user preferences differ in terms of performance, cost, and style.
2. Proposing a heterogeneous GNN-based routing framework that incorporates user, task, query, and LLM nodes to capture latent user preferences.
3. Demonstrating experimentally that PersonalizedRouter outperforms existing baselines (FrugalGPT, RouterDC, HybridLLM, etc.), shows scalability with more users/LLMs, and generalizes to unseen users and unseen LLMs with only few-shot adaptation

**Additional Comments:**

N/A

**Audience:**

Yes

**Audience Explanation:**

Yes, there is interest for at least part of TMLR’s audience:
1. The topic of LLM routing and efficiency is highly relevant to applied ML and system optimization communities.
2. The graph-based personalization angle introduces novelty relative to cost-only routers.
3. However, because the evaluation is fully simulation-driven, the practical impact is less clear. Researchers in TMLR may be interested in the method conceptually, but some will view the lack of real user validation as a serious limitation.

**Broader Impact Concerns:**

Positive: If realized with real data, such routing could significantly reduce costs, personalize user experience, and enable multi-user systems to allocate LLM resources efficiently.
Negative:
1. Fairness & bias: By simulating “user personas” with LLM prompts, the method risks encoding and amplifying stereotypes (e.g., engineer vs. literature enthusiast).
2. Generalization gap: Over-claiming robustness based on synthetic users could mislead practitioners into deploying systems that fail in real-world diversity.
3. Privacy: If deployed on real interaction data, capturing latent user preferences raises concerns about profiling and potential misuse.

**Claims And Evidence:**

Yes

**Claims Explanation:**

1. The experimental results do consistently show gains (up to 16.97% and 9.83% over the best baselines) across two simulation settings, and even larger gains in the large-scale setup ￼.
2. The paper evaluates generalization to new users and LLMs, which is an important practical dimension.
3. Ablation studies (varying GNN depth) and scalability tests are included.

**Requested Changes:**

This paper is basically the same as GraphRouter. The major different part is the simulated user preference, which:
1. is not novel enough by itself. Combining such a module to an existing framework with minor changes doesn't convince me as a major contribution.
2. is not motivated, even far-fetched. GSM8K is a math dataset, evaluated by final correctness scores. Squad contains mainly factoid or short descriptive questions. What roles does user preference play in these types of benchmarks evaluated by correctness? It doesn't make sense.
3. is completely simulated by llm, thus the accuracy and reliability should be a concern (and once again, no idea why the accuracy of a math dataset can be improved by adding user preference... ) And the author didn't provide comparison with GraphRouter or even SOTA baselines on those benchmarks.

---

> ### Author Response · Authors · 2025-09-19
> **Response to Reviewer 2sp7 (Part 1)**
>
> **Q1.It is not novel enough by itself. Combining such a module to an existing framework with minor changes doesn't convince me as a major contribution.**
>
> **Response:** Thanks for your valuable feedback. PersonalizedRouter can extract user preferences directly from interaction data, which is a non-trivial challenge, and thereby supports personalized routing across multiple users. Moreover, PersonalizedRouter and GraphRouter are fundamentally distinct in their application scenarios. GraphRouter performs LLM routing based on fixed cost/performance weights, and is therefore limited to fixed-user settings. As we pointed out in our paper (Section 1, Introduction, second paragraph):
>
> >“GraphRouter constructs a heterogeneous graph based on user interaction data and uses a Graph Neural Network (GNN) to predict the most suitable LLM. However, a fundamental limitation inherent in current approaches is that they fail to adequately account for user preferences (Table 1).”
>
> In contrast, PersonalizedRouter is explicitly designed for multi-user scenarios. In our framework, user nodes are introduced into the graph, and edges encode different types of user preferences (e.g., cost–performance trade-offs, stylistic preferences). Consequently, this enables PersonalizedRouter to learn latent user preferences directly from interaction data and achieve personalized LLM selection for multiple users. To the best of our knowledge, this aspect has not been explicitly addressed by prior methods.
>
> **Q2.It is not motivated, even far-fetched. GSM8K is a math dataset, evaluated by final correctness scores. Squad contains mainly factoid or short descriptive questions. What roles does user preference play in these types of benchmarks evaluated by correctness? It doesn't make sense.**
>
> **Response:** Thanks for your valuable questions and insightful feedback. We answer your questions in two parts.
>
> **[It is not motivated, even far-fetched.]** Our motivation is to support personalized LLM selection in multi-user routing scenarios that account for user preferences in cost/performance trade-offs or response styles.
>
> **[What roles does user preference play in these types of benchmarks evaluated by correctness?]** Although correctness is the primary evaluation metric in GSM8K, in practical scenarios, users’ stylistic preferences also influence LLM choice. As stated in the introduction of the paper:
>
> >In addition to differences in response quality and cost, LLMs also exhibit diverse response styles, which influence users’ understanding of the query.
>
> For instance, while some LLMs provide detailed step-by-step reasoning, others present only the key idea and final result. Because users hold distinct preferences regarding approaches to solving math problems, these preferences ultimately affect the final decision. Similarly, in SQuAD, some LLMs give brief answers while others enrich their responses with additional evidence. Detailed cases of GSM8K and SQuAD are presented in Appendix A.7 of the latest revised version. Therefore, both GSM8K and SQuAD are useful in evaluating not only cost/performance trade-offs but also response-style preferences. Experimental results confirm that PersonalizedRouter is effective in both aspects: ensuring correctness under cost/performance constraints and capturing user stylistic preferences in LLM judge scenarios.
>
> **Q3.It is completely simulated by llm, thus the accuracy and reliability should be a concern (and once again, no idea why the accuracy of a math dataset can be improved by adding user preference... ) And the author didn't provide comparison with GraphRouter or even SOTA baselines on those benchmarks.**
>
> **Response:**  Thanks for the reviewer’s insightful question. We will answer your questions one by one.
>
> **[Accuracy and reliability concerns of LLM-based simulation]:**  Since collecting diverse and large-scale real user preference data across multiple LLMs is infeasible, we adopt LLM-simulated preferences. We think that such a simulation is valuable for validating whether the router can capture latent user preferences, and prior studies ([1],[2]) have also treated LLM simulation as a reliable methodology. To further reduce potential bias from relying on a single LLM as a judge, we conducted additional experiments with 3 LLMs as judges, each combined with two types of instruction prompts. The experimental results are shown in Table 1.
>
> **[1]** Zhang, Z., Liu, S., Liu, Z., Zhong, R., Cai, Q., Zhao, X., Zhang, C., Liu, Q., & Jiang, P. (2025). LLM-Powered User Simulator for Recommender System. Proceedings of the AAAI Conference on Artificial Intelligence, 39(12), 13339-13347.
>
> **[2]** Lixi Zhu, Xiaowen Huang, and Jitao Sang. 2025. A LLM-based Controllable, Scalable, Human-Involved User Simulator Framework for Conversational Recommender Systems. In Proceedings of the ACM on Web Conference 2025 (WWW '25). Association for Computing Machinery, New York, NY, USA, 4653–4661.

---

> ### Author Response · Authors · 2025-09-19
> **Response to Reviewer 2sp7 (Part 2)**
>
> **Table1: Comparison between different methods under two simulation strategies in the large-scale experimental setting. The Deltas represent the improvement of the Reward over the best baseline. The Reduction indicates the relative reduction in time cost compared to the baseline with the highest time cost.**
>
>
> | Scenario    | multi-cost-efficiency simulation             | LLM-as-a-Judge strategy                |
> |-------------|----------------------------------------------|----------------------------------------|
> | Method      | Reward &nbsp;  Delta (%) &nbsp;  Time   &nbsp;  Reduction (%)   | Accuracy &nbsp;  Delta (%) &nbsp;  Time   &nbsp; Reduction (%) |
> | HybridLLM   | 0.119  &emsp;  -43.33    &ensp;&emsp;  4:32   &emsp;&emsp;  98.50  &nbsp;         | 0.196    &emsp;&emsp;&emsp;  0.00      &emsp;&emsp;  4:56   &emsp;&nbsp; 98.40         |
> | FrugalGPT   | 0.141  &emsp;  -32.86    &ensp;&emsp;  47:44  &emsp;&emsp;  84.29  &nbsp;         | 0.108    &emsp;&emsp; -44.90     &emsp;&emsp;  49:31  &ensp;&nbsp; 83.92         |
> | GraphRouter | 0.204  &emsp;  -2.86     &ensp;&emsp;&ensp;  8:26   &emsp;&emsp;  97.22  &nbsp;         | 0.137    &emsp;&emsp; -30.10     &emsp;&emsp;  9:28   &emsp;&ensp; 96.93         |
> |RouterDC | 0.210  &nbsp;&emsp; 0.00     &ensp;&emsp;  303:47 &emsp;&emsp;  0.00   &nbsp;         | 0.153    &emsp;&emsp; -21.94     &emsp;&emsp;  308:02 &ensp;&nbsp; 0.00          |
> | Ours        | 0.244  &emsp;   16.19    &ensp;&emsp; 10:15   &emsp;&emsp; 96.63   &nbsp;         | 0.313    &emsp;&emsp;&nbsp;  59.69     &emsp;&emsp;  11:37  &emsp;&nbsp; 96.23         |
> | Oracle      | 0.310  &emsp;   47.62    &ensp;&emsp;  /      &emsp;&emsp;  /      &nbsp;         | 1.000    &emsp;&emsp; 410.20     &emsp;&emsp;  /      &ensp;&nbsp; /             |
>
> Moreover, following the constructive suggestion, we additionally collected a small-scale real-user dataset consisting of 10 LLMs, 40 users, and 80 queries. The experimental results are shown in Table 2. These experiments confirm that PersonalizedRouter generalizes beyond simulated settings and is also effective in real-user scenarios.
>
> **Table2:Comparison between different methods under small-scale human interaction dataset. The Improvement is measured relative to the best baseline.**
>
> | Scenario    |       LLM-as-a-Judge strategy      |
> |-------------|------------------------------------|
> | Method      |       Accuracy &nbsp;  Improvement (%)   |
> | HybridLLM   |       0.315  &emsp;&emsp;&ensp;   -23.73             |
> | FrugalGPT   |       0.350  &emsp;&emsp;&ensp;   -15.25             |
> | GraphRouter |       0.357  &emsp;&emsp;&ensp;   -13.56             |
> | RouterDC    |       0.413  &emsp;&emsp;&ensp;&nbsp;    0.00              |
> | PersonalizedRouter| 0.438  &emsp;&emsp;&ensp;&nbsp;    6.05              |
> | Oracle      |       1.000  &emsp;&emsp;&ensp;   142.13             |
>
> **[why the accuracy of a math dataset can be improved by adding user preference]:**  We would like to clarify that our results do not reflect an improvement in the accuracy of math datasets through the addition of user preferences. Instead, under the LLM-as-a-Judge strategy, the judge selects the response that best aligns with its stylistic preference as the best answer (Section 4.2). Accordingly, our contribution lies in improving the router’s ability to select the best answer judged by the LLM judge, rather than increasing task-level accuracy on the math dataset.
>
> **[And the author didn't provide comparison with GraphRouter or even SOTA baselines on those benchmarks.]:** In the latest revision, we have incorporated GraphRouter into all experiments and conducted comparisons with other baselines. The experimental results are reported in Tables 1, 2, 3, 4, and 5. Our PersonalizedRouter still outperforms all other baselines.
>
> **Q4.Fairness & bias: By simulating “user personas” with LLM prompts, the method risks encoding and amplifying stereotypes (e.g., engineer vs. literature enthusiast).**
>
> **Response:**  We thank the reviewer for raising the above concerns. Our work primarily focuses on demonstrating that PersonalizedRouter can effectively learn user preferences from interaction data and achieve personalized LLM selection. The issue of whether simulating users with LLMs may introduce stereotypes is indeed an important open question, but it is not the central focus of our work. As clarified in response to Q3, LLM simulation has been adopted in prior work and provides meaningful value for validating whether a router can capture latent user preferences. Furthermore, to alleviate concerns about potential bias from relying on a single LLM to simulate users, we conducted additional LLM-judge experiments using three LLMs, each with two distinct instruction prompts. The experimental results are presented in Table 1.

---

> ### Author Response · Authors · 2025-09-19
> **Response to Reviewer 2sp7 (Part 3)**
>
> **Q5.Generalization gap: Over-claiming robustness based on synthetic users could mislead practitioners into deploying systems that fail in real-world diversity.**
>
> **Response:** Thanks for raising the above concerns. As mentioned in our response to Q3, prior work has also adopted simulated data to get user preferences as a reliable methodology. In particular, LLM-based simulation can mimic latent preferences of real users, thereby providing a strong reference value for validating whether a router is capable of capturing such hidden preferences. Moreover, we collected a small-scale real-user interaction dataset to evaluate the generalization of the models. The results reported in Table 1 further confirm that our approach is effective not only in simulated environments but also in real-user scenarios.
>
> **Q6.Privacy: If deployed on real interaction data, capturing latent user preferences raises concerns about profiling and potential misuse.**
>
> **Response:** Thanks for the reviewer’s insightful questions. It is worth noting that user profiles are only used as system prompts for LLMs to simulate users. During actual training and inference, the model does not observe these profiles. PersonalizedRouter does not rely on accessing sensitive user profiles. Instead, it learns preferences from query–LLM interaction data. Consequently, the embedding of a user node is not a direct representation of the user’s real profile, but rather an abstract latent representation designed to better predict the most suitable LLM for a given query.
>
>
> **Table3: Comparison between different methods under two simulation strategies. The Improvement is measured relative to the best baseline.**
>
> | Scenario&nbsp; | Multi-cost-efficiency Simulation      | LLM-as-a-Judge    |
> |----------------|---|-------------------|
> | Method     | Reward &nbsp;  Improvement (%) | Accuracy &nbsp;  Improvement (%)  |
> | HybridLLM  | 0.141  &emsp;&emsp;&emsp; -36.20  | 0.347  &emsp;&emsp;&emsp;   -14.74            |
> | FrugalGPT  | 0.218  &emsp;&emsp;&emsp; -1.36  | 0.354  &emsp;&emsp;&emsp;   -13.22            |
> | GraphRouter| 0.221  &emsp;&emsp;&emsp; 0.00    | 0.364  &emsp;&emsp;&emsp;   -10.57            |
> | RouterDC   | 0.208  &emsp;&emsp;&emsp; -5.88   | 0.407  &emsp;&emsp;&emsp;   0.00              |
> | Ours       | 0.255  &emsp;&emsp;&emsp; 15.38   | 0.447  &emsp;&emsp;&emsp;   9.83              |
> | Oracle     | 0.304  &emsp;&emsp;&emsp; 37.56  | 1.000  &emsp;&emsp;&emsp;   145.70            |
>
> **Table4: Comparison between different methods under simulation strategies in the new user experimental setting. The Improvement is measured relative to the best baseline.**
>
> | Scenario         | multi-cost-efficiency simulation             | LLM-as-a-Judge strategy                |
> |-------------     |----------------------------------------------|----------------------------------------|
> | Method           | Reward  &nbsp; Improvement (%) | Accuracy   Improvement (%)  |
> | HybridLLM        | -0.142   &emsp;&emsp;&emsp;   -240.59         | 0.294  &emsp;&emsp;&emsp;&ensp;    0.00             |
> | FrugalGPT        | 0.044    &emsp;&emsp;&emsp;   -56.44          | 0.192  &emsp;&emsp;&emsp;   -34.69            |
> | GraphRouter      | 0.083    &emsp;&emsp;&emsp;   -17.82          | 0.256  &emsp;&emsp;&emsp;   -12.93            |
> | RouterDC         | 0.101    &emsp;&emsp;&emsp;&ensp;    0.00           | 0.208  &emsp;&emsp;&emsp;   -29.25            |
> | Ours (Few-shots) | 0.070    &emsp;&emsp;&emsp;  -30.69           | 0.313  &emsp;&emsp;&emsp;&ensp;    6.46             |
> | Ours (Trained)   | 0.108    &emsp;&emsp;&emsp;&ensp;   6.93            | 0.326  &emsp;&emsp;&emsp;&ensp;   10.88             |
> | Oracle           | 0.116    &emsp;&emsp;&emsp;&ensp;   14.85           | 1.000  &emsp;&emsp;&emsp;&ensp;   240.14            |
>
> **Table5: Comparison between different methods under simulation strategies in the new LLM experimental setting. The Improvement is measured relative to the best baseline.**
>
> | Scenario        | multi-cost-efficiency simulation             | LLM-as-a-Judge strategy                |
> |-------------    |----------------------------------------------|----------------------------------------|
> | Method          | Reward &nbsp;  Improvement (%) | Accuracy &nbsp;  Improvement (%)  |
> | HybridLLM       | 0.137 &emsp;&emsp;&emsp;  -37.16   | 0.273  &emsp;&emsp;&emsp;    0.00             |
> | FrugalGPT       | 0.182 &emsp;&emsp;&emsp;  -16.51   | 0.038  &emsp;&emsp;&emsp;   -86.08     |
> | GraphRouter     | 0.203 &emsp;&emsp;&emsp;  -6.88    | 0.247  &emsp;&emsp;&emsp;   -9.52    |
> | RouterDC        | 0.218 &emsp;&emsp;&emsp;   0.00    | 0.199  &emsp;&emsp;&emsp;  -27.10     |
> | Ours (few-shot) | 0.201 &emsp;&emsp;&emsp;  -7.80   | 0.329  &emsp;&emsp;&emsp;   20.51     |
> | Ours (trained)  | 0.234 &emsp;&emsp;&emsp;   7.34    | 0.383  &emsp;&emsp;&emsp;   40.29    |
> | Oracle          | 0.371 &emsp;&emsp;&emsp;  70.18      | 1.000  &emsp;&emsp;&emsp;  266.30    |

---

### Author Response · Authors · 2025-09-19
**General Response**

We sincerely thank the reviewers and editors for their detailed and constructive feedback. In response to the issues raised, we have revised the manuscript, with the modified parts highlighted in blue. According to reviewer h7mj’s request, we expanded the dataset from 10 users to 1,000 users to verify the feasibility of our model in larger-scale user scenarios. Following reviewer iphw’s suggestion, we collected a small-scale real-user dataset and conducted experiments. According to the requests of reviewers 2sp7 and h7mj, we added GraphRouter as a baseline and supplemented the corresponding experiments. In addition, we added to the appendix the role of the GSM8K and SQuAD datasets in user-preference experiments, the t-SNE visualization figure, and the user questionnaire template. We also refined the detailed descriptions of tables.

We hope that these revisions not only enhance the completeness of the paper but also address both major and minor concerns raised by the reviewers. We hope these revisions will improve the evaluation of our submission. Thank you for your thoughtful consideration.

---

### Decision · Action_Editor_oYR7 · 2025-10-20

**Recommendation:** Accept with minor revision

**Additional Comments:**

The authors have done an outstanding job of addressing the reviewers' initial, and very valid, concerns. The revised manuscript is substantially stronger than the original submission. By adding large-scale experiments, a real-user study, and the critical GraphRouter baseline, the authors have provided the necessary evidence to support their claims. Reviewers h7mj and iphw both confirmed their concerns were met. The paper now clearly meets the TMLR acceptance criteria.

[Requested Revision]
Please revise Section 5.5, "Comparison with Baselines on Real-User Dataset," to more clearly describe the evaluation protocol. Specifically, please clarify how the collected human preferences (where users selected their single favorite response ) were used for scoring and how this setup relates to the "LLM-as-a-Judge" scenario referenced in Table 8.

**Audience:**

Yes

**Audience Explanation:**

All three reviewers unanimously agreed that the topic is of clear interest to the TMLR audience. The problem of selecting an appropriate LLM from a rapidly growing and diverse set of options is a highly relevant and practical challenge for the community.

**Claims And Evidence:**

Yes

**Claims Explanation:**

The initial submission had significant gaps between its claims and its evidence, which were rightly identified by the reviewers. Key concerns included (1) a very small experimental scale of only 9 users, (2) a complete reliance on simulated data without real-user validation, and (3) missing the GraphRouter baseline.

The authors provide a comprehensive response to these critiques. In their revisions, they have:

1. Expanded Scale: Conducted new large-scale experiments, increasing the user count from 9 to 1,000.
2. Added Real-User Data: Collected and tested on a new, small-scale dataset from 40 real human users, addressing the concern about relying purely on simulation.
3. Included Key Baselines: Added GraphRouter as a baseline to all relevant experiments and clarified the methodological distinction of using user nodes for personalization.
4. Strengthened Evaluation: Mitigated potential bias in the LLM-as-a-Judge strategy by using three different LLM judges.
5. Diversified Models: Updated the candidate LLM pool to include recent and domain-specific models.

These comprehensive additions directly address the major concerns.

---

> ### Author Response · Authors · 2025-11-12
> **Official Comment by Authors**
>
> Thank you for your suggestions. We have provided detailed responses and made corresponding revisions to the manuscript.
>
> 1. In Section 5.5, we have clarified how user selections are used as the evaluation protocol.
> 2. To avoid misunderstanding, we have also revised the wording in Section 5.5, replacing LLM-as-a-Judge with Human-as-a-Judge, since the best-answer labels were determined by human.
>
> We sincerely thank you and all reviewers for your valuable feedback and support. Please let us know if you have any further suggestions.